# THEORETICAL MODELING OF LARGE LANGUAGE MODEL SELF-IMPROVEMENT TRAINING DYNAMICS THROUGH SOLVER-VERIFIER GAP

**Yifan Sun**[*], **Yushan Liang**[*], **Zhen Zhang, Xin Liu**[†] **& Jiaye Teng**[†]
School of Statistics and Data Science
Shanghai University of Finance and Economics
Shanghai, China
{yifan.sun, yushanl}@stu.sufe.edu.cn, tengjiaye@sufe.edu.cn

## ABSTRACT

Self-improvement is a significant techniques within the realm of large language model (LLM), aiming to enhance the LLM performance without relying on external data. Despite its significance, generally how LLM performance evolves during the self-improvement process remains underexplored. In this paper, we theoretically model the training dynamics of self-improvement via the concept of *solver-verifier gap*. This is inspired by the conjecture that the performance enhancement of self-improvement stems from the gap between LLM's solver capability and verifier capability. Based on the theoretical framework, we further show how to model the entire training trajectory. This framework allows quantifying the capability limit of self-improvement by fitting the theoretical model to the experiment results. We validate the effectiveness of the theoretical framework on various LLMs and datasets. Beyond self-improvement, we extend our analysis to investigate how external data influences these dynamics within the framework. Notably, we find that under limited external data regimes, such external data can be utilized at any stage without significantly affecting final performances, which accords with the empirical observations.

## 1 INTRODUCTION

Large language models (LLMs) have emerged as one of the most pivotal frontiers in artificial intelligence, propelling the development of diverse applications such as chatbots (Brown et al., 2020), mathematical reasoning (Wei et al., 2022; Shao et al., 2024), and robotics (Wu et al., 2023). Despite their remarkable success, the training of LLMs typically necessitates massive data. In practice, data collection often confronts significant challenges, and there are even concerns that available data sources could be depleted in the future (Villalobos et al., 2024; Shen et al., 2025; Muennighoff et al., 2023; Wang et al., 2024a). This data bottleneck motivates researchers to explore alternative training strategies (Gao et al., 2020; Dong et al., 2024).

Among these strategies, a growing body of work focuses on training or fine-tuning LLMs using the data they generate, a process known as self-improvement (Bai et al., 2022; Huang et al., 2023; Wang et al., 2023; Pang et al., 2024). Self-improvement methodologies initiate with a pre-trained LLM, utilize the model to generate new data, and then fine-tune the model with the generated data. Empirical studies have shown that this approach yields promising results across various domains (Zelikman et al., 2022; Wang et al., 2023; Tian et al., 2024). However, the theoretical underpinnings of self-improvement remain under-explored (Song et al., 2025; Huang et al., 2025). Specifically, there is a lack of theoretical models to explain its mechanisms and insufficient evidence to fully understand the training dynamics involved in this process.

---

[*]Equal contribution.
[†]Corresponding author

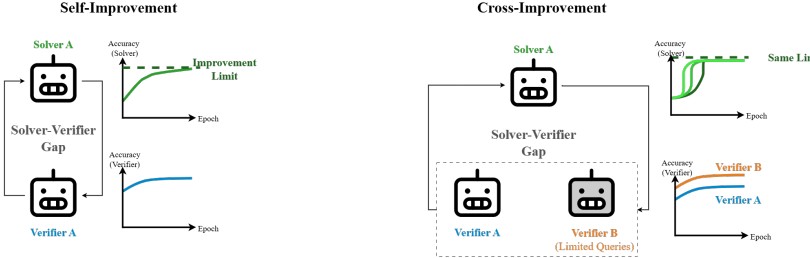

Figure 1: Illustration of the theoretical results. The training dynamics are potentially empowered by the solver-verifier gap under the theoretical framework. For self-improvement (left), the accuracy exhibits exponential curves towards a limit. For cross-improvement (right), adding external data (a different verifier with limited queries) at different times yields similar performance.

In this paper, we theoretically model the training dynamics of LLM self-improvement, inspired by the conjecture that self-improvement capability arises from the gap between an LLM's solver capability and verifier capability [1]. Specifically, we define these two capabilities as follows:

- Solver capability $U_s(t)$: The quality of responses directly generated from LLM;
- Verifier capability $U_v(t)$: The quality of responses generated from LLM and evaluated by LLM.

In practice, different metrics can be used to quantify these capabilities. For tasks with ground-truth labels, the $0-1$ training loss can serve as a direct measure. For tasks without ground truth, uncertainty quantification can be adopted, as it correlates strongly with model capability (Huang et al., 2025). Based on the above discussions, we model the training dynamics of LLM self-improvement using the coupled differential equations in Equation 1, inspired by *potential energy* in physics (Rankine, 1853):

$$\frac{dU_s(t)}{dt} = -\alpha E(t), \quad \frac{dU_v(t)}{dt} = -\beta E(t), \tag{1}$$

where $\alpha, \beta$ denote the coefficients, $t$ denotes the epoch, $U_s(t)$ denotes the solver capability, $U_v(t)$ denotes the verifier capability, and $E(t)$ denotes the capability gap related to $U_s(t) - U_v(t)$. We omit the initial conditions for simplicity. Under such a framework, the resulting dynamics would be

$$U_s(t) \approx \alpha' e^{-k(\alpha-\beta)t} + U_{s,\infty}, \quad U_v(t) \approx \beta' e^{-k(\alpha-\beta)t} + U_{v,\infty}, \tag{2}$$

where $\alpha', \beta', k$ denotes the constants with detailed formulations in Section 3.2, and $U_{s,\infty}, U_{v,\infty}$ represent the solver capability and the verifier capability at convergence, respectively. Along the trajectory, both capabilities follow exponential convergence laws according to the framework. Besides, we specifically focus on the solver's ultimate capability $U_{s,\infty} = \frac{1}{\alpha-\beta}(\alpha U_{v,0} - \beta U_{s,0} + \alpha \frac{b}{k})$. This result reveals that the solver's ultimate capability relates to both the initial capability $U_{s,0}, U_{v,0}$ and the coefficient $\alpha, \beta$. Therefore, the ultimate capability might be improved given a larger verifier-solver gap at initialization, in accordance with our insights. Detailed derivations are provided in Section 3.2.

From the experimental perspective, we find in Section 4 that such theoretical modeling demonstrates strong practical efficacy. Specifically, experimental results across various models and datasets in Figure 3 reveal that the capability dynamics indeed follow an exponential law, as implied by the theoretical framework in Section 3. Besides, we observe in Figure 3 that the verifier capability consistently outperforms the solver throughout the self-improvement process, which might be empirical evidence of the solver-verifier gap's crucial role in driving capability improvement. To further investigate the role of the solver-verifier gap, we conduct experiments in Section 4.3 on multiple LLMs demonstrating that the gap generally occurs in practice. These findings hold across varying sample sizes and even under cross-evaluation scenarios.

We further analyze in Section 5 the application of this framework in cross-improvement, a potential approach to enhance the ultimate capability of self-improvement. Cross-improvement in this paper refers to the utilization of external data in the verification step (details in Figure 7). Within the

---

[1]We term this the solver-verifier gap to ensure consistency with our mathematical notation ($U_s$ for Solver, $U_v$ for Verifier) and our specific Best-of-N implementation. We explicitly acknowledge that this concept builds directly upon the generation-verification gap originally conceptualized by Song et al. (2025), and we adopt this terminology with deep respect for their foundational contribution.

theoretical framework, we contend that cross-improvement outperforms self-improvement due to the enhanced verification capabilities. We then derive the training dynamics under the cross-improvement regimes, with enhanced verification capabilities compared to self-improvement. This analysis can further assist in answering the question: given the limited amount of external data, how should one allocate these external data during the cross-improvement training process? Our theoretical analyses demonstrate that the timing of using external data is not crucial; rather, the utilization of such data genuinely enhances the training process. Therefore, external data can be incorporated at any stage as desired. Experiments in Section 5.2 on the cross-improvement validate the theoretical findings. Our main contributions are presented in Appendix A.2.

## 2 RELATED WORK

**Self-Improvement.** Self-improvement aims to enhance model performance without relying on external information (Huang et al., 2023; Wang et al., 2023; Bai et al., 2022; Pang et al., 2024). Self-improvement is of significant importance as it enables models to adapt and evolve autonomously, thereby facilitating their effectiveness across a wide range of real-world scenarios, such as reasoning (Zelikman et al., 2022; Peng et al., 2024; Huang et al., 2023), alignment (Wang et al., 2023; 2024b; Ding et al., 2024), and planning (Tian et al., 2024). In this paper, we consider a branch of self-improvement that fine-tunes with the output of LLM (Amini et al., 2024; Sessa et al., 2024; Gui et al., 2024; Pace et al., 2024; Ouyang et al., 2022; Rafailov et al., 2023). Alternative approaches to self-improvement include self-distillation (Buciluǎ et al., 2006; Hinton et al., 2015; Zhang et al., 2019) which involves transferring knowledge from a larger, more complex model to a smaller one, and self-correction (Kumar et al., 2024; Liu et al., 2024) where the model identifies and rectifies its own errors, *etc*. A body of research has also explored the potential negative consequences of self-improvement, including degradation issues (Bertrand et al., 2024; Gerstgrasser et al., 2024) and failures on out-of-domain reasoning tasks (Yuan et al., 2025).

**Theoretical Understandings on Self-Improvement.** Theoretical insights into self-improvement could potentially enhance comprehension, thereby making self-improvement more reliable (Yampolskiy, 2015). Previous works have theoretically studied self-improvement via self-distillation techniques, providing convergence rates for linear models (Mobahi et al., 2020; Frei et al., 2022; Das & Sanghavi, 2023; Pareek et al., 2024), neural networks (Allen-Zhu & Li, 2023), and general models (Boix-Adsera, 2024). In the realm of LLM, several works have theoretically explored self-improvement with in-context alignment (Wang et al., 2024c), reinforcement learning (Talvitie, 2017; Choi et al., 2024; Gandhi et al., 2025), meta learning (Kirsch & Schmidhuber, 2022), and diffusion models (Fu et al., 2024). Nevertheless, the theoretical understanding of self-improvement in the context of LLM training dynamics remains underexplored.

Most relevant to our work is Song et al. (2025) and Huang et al. (2025). Song et al. (2025) posits that the key to self-improvement lies in the generation-verification gap and further examines the relationship between this gap and pre-training flops. Huang et al. (2025) further posits that the improvement stems from a sharpening mechanism, in which the verification step sharpens the model performance on the high-quality sequences. Our paper draws inspiration from Song et al. (2025); Huang et al. (2025), as we employ the concept of a solver-verification sharpening gap in the theoretical analysis, and the training policy in this paper follows Huang et al. (2025). However, different from Song et al. (2025); Huang et al. (2025), our research primarily centers on developing the self-improvement *dynamics* based on the solver-verification sharpening gap. Additionally, we delve deeper into understanding how cross-improvement works within this framework.

**Cross-Improvement.** Besides self-improvement approaches, a branch of papers focuses on enhancing the capabilities of LLM through external data, namely, cross-improvement. One of the most frequently utilized sources of external data is human-annotated data (Ouyang et al., 2022; Lightman et al., 2024; Borchers et al., 2025). Despite its utility, the collection of such data is extremely resource-intensive. Moreover, relying solely on human-annotated data restricts the potential for LLM to surpass human performance. Another potential source of external data stems from stronger models (Ho et al., 2023; Chang et al., 2023; Lee et al., 2024), while access to these stronger models often presents significant challenges. Our paper also considers the scenario of cross-improvement that leverages a limited number of tokens from stronger models.

# 3 THEORETICAL MODELING OF SELF-IMPROVEMENT TRAINING DYNAMICS

This section presents a theoretical framework for sketching training dynamics in LLM self-improvement. We start by introducing necessary notations of self-improvement in Section 3.1. We then theoretically model the self-improvement dynamics in Section 3.2.

## 3.1 SOLVER AND VERIFIER

This section introduces the basic notations and definitions, as well as the definitions of solver capability and verifier capability. We start from the basic notations on data and models.

**Notations.** Let $(x, y)$ denote a prompt-response pair, with $y^{[k]}$ denoting the $k$-th token and $y^{[1:k]}$ denoting the first $k$ tokens. We use $L(y)$ to represent the length of $y$. Let $\pi_f(y|x)$ denote the probability that a model $f$ generates a response $y$ given a prompt $x$, where $\pi_f(y|x)$ can be split with the auto-regressive structure of the response $\pi_f(y|x) = \prod_{k=1}^{L(y)} \pi_f(y^{[k]}|y^{[1:k-1]}, x)$. We denote the *best* response as $y^*$, that is, the ground truth response. Ideally, a model's performance could be measured by its loss relative to this ground truth, for instance, $L_f(\hat{y}) = \|y^* - \hat{y}\|$. However, as not all tasks have an accessible ground truth, a different metric is required. Therefore, we also use the uncertainty metric following Huang et al. (2025) in our framework. We define the uncertainty for a response $\hat{y}$ given its prompt $x$ and a model $f$ as its negative log-likelihood: $U_f(\hat{y}) = -\log \pi_f(\hat{y}|x)$. The model's capability is inversely related to the uncertainty: a lower uncertainty signifies a higher capability. Further discussion is available in Appendix A.3.

**Solver.** The solver is regarded as the model capability to return responses with low uncertainty. Therefore, for each prompt $x_i$, we sample one response $\hat{y}_i(t)$ to represent the solver solution. That is,

$$\hat{y}_i \sim \pi_f(\cdot|x). \tag{3}$$

Note that we only draw one response for each prompt due to calculation efficiency. An alternative solution is to generate multiple responses and use the whole distribution, but the two policies are similar since we already take average operations over (*i.i.d.*) prompts.

**Verifier.** We use LLM itself as the verifier. For each prompt $x_i$, we first sample $N$ responses based on the LLM output $\hat{y}_{i,1}, \cdots, \hat{y}_{i,N} \sim \pi_f(\cdot|x_i)$. We then ask the LLM to evaluate these responses with a score $s(\hat{y}_{i,j}) \in [0, 1]$. The Best-of-N (BoN) response is then defined as

$$\hat{y}_i^{\text{BoN}} = \underset{\{\hat{y}_{i,j}:s(\hat{y}_{i,j}) \geq \sigma\}}{\arg\min} \frac{1}{L(\hat{y}_{i,j})} U_f(\hat{y}_{i,j}|x_i), \tag{4}$$

where $\sigma$ denotes the threshold parameter, and we use $1/L(\hat{y}_{i,j})$ as a regularizer to discourage those short responses. The BoN solution first eliminates those solutions with small scores $s(\hat{y}_{i,j})$, and then finds the solution with the best capability. We deploy such a mixed strategy to enhance the computational stability; as a comparison, a strategy with only score $s(\hat{y}_{i,j})$ might cause $\hat{y}_i^{\text{BoN}}$ to have large variance. Obviously, the BoN solution merges the LLM verifier capability and the LLM output. Therefore, the verifier capability can be calculated based on the uncertainty of the BoN solution. We finally remark that our paper uses a slightly different BoN policy compared to Huang et al. (2025) where they do not eliminate those responses with low scores, since we want to include more verification capability in the BoN response.

**Uncertainty Metrics.** Based on the above discussion, we use the average uncertainty of LLM response $\hat{y}$ to represent the solver capacity and use the average uncertainty of BoN response $\hat{y}^{BoN}$ to represent the verification capability. Therefore, we define the *solver uncertainty* $U_s(t)$ and *verifier uncertainty* $U_v(t)$ as

$$U_s(t) \triangleq -\frac{1}{n}\sum_{i=1}^{n} \log \pi_f(\hat{y}_i(t)|x_i), \quad U_v(t) \triangleq -\frac{1}{n}\sum_{i=1}^{n} \log \pi_f(\hat{y}_i^{BoN}(t)|x_i), \tag{5}$$

where $\hat{y}_i(t)$ denotes the LLM output. In our framework, uncertainty metrics serve as inverse metrics of capability: a lower uncertainty value ($U_s$ or $U_v$) indicates a higher corresponding capability. Note that both $U_s(t)$ and $U_v(t)$ contains randomness, since $\hat{y}_i(t)$ is randomly generated by LLM, and the score $s(\cdot)$ in $\hat{y}_i^{\text{BoN}}(t)$ is also randomly generated by LLM. However, since the prompts $x_i$ are independent, the randomness could be controlled when the number of prompts $n$ is large.

**Self-Improvement.** We deploy self-improvement based on the above solver-verifier framework, similar to Huang et al. (2025). Notably, the verifier is slightly different, since we want to include more verification information. Overall, we first generate responses from the LLM. The optimization objective is to minimize the average uncertainty of BoN responses. The loss function $L_t(f)$ for a training step $t$ with function $f$ is defined as the verifier uncertainty $U_v(t)$: $L_t(f) \triangleq U_v(t) = -\frac{1}{n}\sum_{i=1}^{n} \log \pi_f(\hat{y}_i^{\text{BoN}}(t)|x_i)$. By minimizing $L_t(f)$, we steer the model to increase the likelihood of generating high-quality responses, improving its solver capability. We summarize one-step self-improvement algorithm in Algorithm 1.

## 3.2 Self-Improvement Dynamics

In this section, we aim to analyze the dynamics of self-improvement. Given the immense complexity of the underlying neural network processes, deriving dynamics from first principles is currently intractable. Therefore, we adopt a phenomenological modeling approach, which is common in the study of complex systems. Our techniques are inspired by the concept of potential energy, a widely used concept in physics (Rankine, 1853). Following Huang et al. (2025) and Song et al. (2025), we argue that the self-improvement comes from the *Capability Gap* $G(\cdot)$, defined as the gap between $U_s(t)$ and $U_v(t)$, namely,

$$G(t) \triangleq U_s(t) - U_v(t) = -\frac{1}{n}\sum_{i=1}^{n} \log \frac{\pi_f(\hat{y}_i(t)|x_i)}{\pi_f(\hat{y}_i^{BoN}(t)|x_i)}. \tag{6}$$

We assume that the change of the solver and verifier capability is driven by a *gap potential energy* $E(t)$, expressed as a function of the Capability Gap $G(t)$ (Equation (6)), namely $E(t) = f(G(t))$, where $f(G)$ is a differentiable and monotonically increasing function for $G \geq 0$, with its minimum $f(0) = 0$. In this paper, we simply use uncertainty to evaluate the response quality, leading the capability gap defined with the uncertainty, as discussed in Section 3.1. We assume that the solver capability and the verifier capability both increase during the process of self-improvement, which is widely observed in the related works (Song et al., 2025). With the analysis above, following the concept of potential energy, the theoretical framework starts with the following assumptions:

$$U_s(t)|_{t=0} = U_{s,0}, \qquad U_v(t)|_{t=0} = U_{v,0}, \tag{7}$$

$$\frac{dU_s(t)}{dt} = -\alpha E(t), \quad \frac{dU_v(t)}{dt} = -\beta E(t), \tag{8}$$

where $\alpha, \beta \geq 0$ are coefficients related to the decreasing rate of $U_s(t), U_v(t)$. Equation (7) represents the initial conditions. Since for LLM, the verification capability usually outperforms the solver capability in applications, we assume that $U_{s,0} > U_{v,0}$. Equation (8) represents the iterative conditions, where we assume that $\alpha > \beta$, indicating that the solver capability increases faster than the verifier. Based on the above assumptions, we derive the differential equation governing the Capability Gap:

$$\frac{dG(t)}{dt} = -(\alpha - \beta)E(t). \tag{9}$$

However, for a non-linear function $f(G)$, a closed-form solution for above integral is not guaranteed. To simplify this problem, we approximate the potential energy function $E(t)$ with a linear form, $E(t) \approx kG(t) - b$, derived from a first-order Taylor expansion. Figure 2 illustrates the strong linear relationship between uncertainty gap $G$ and its rate of change $dG/dt = -(\alpha - \beta)E(t)$ on Phi-4-mini with QE method, which provides robust evidence for our approximation. Experiment results on other models are presented in Figure 8. Based on the Equation (8), we derive Proposition 3.1:

**Proposition 3.1** *Assume $k(\alpha - \beta) > 0$, the dynamics of the potential energy $E(t)$, the capability gap $G(t)$. Let the potential energy function $f(G)$ be linearly approximated around an initial state $G_0$ by its tangent line $f(G) \approx kG - b$, where $k = f'(G_0)$ and $b = f'(G_0)G_0 - f(G_0)$. Capabilities $U_s(t)$ and $U_v(t)$ are governed by the following exponential decay functions:*

$$E(t) \approx k\delta e^{-k(\alpha-\beta)t}, \qquad G(t) \approx \delta e^{-k(\alpha-\beta)t} + G_\infty, \tag{10}$$

$$U_s(t) \approx \alpha' e^{-k(\alpha-\beta)t} + U_{s,\infty}, \quad U_v(t) \approx \beta' e^{-k(\alpha-\beta)t} + U_{v,\infty}, \tag{11}$$

*where the coefficients are defined as:* $\delta = U_{s,0} - U_{v,0} - \frac{b}{k}, \alpha' = \frac{\alpha\delta}{\alpha-\beta}, \beta' = \frac{\beta\delta}{\alpha-\beta}$, *and the values at convergence* $(t \to \infty)$ *are given by:* $G_\infty = \frac{b}{k}, U_{s,\infty} = U_{s,0} - \alpha', U_{v,\infty} = U_{v,0} - \beta'$.

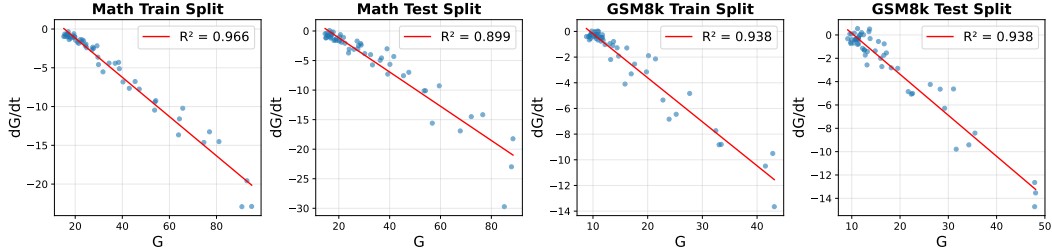

Figure 2: Verification on linear relationship between the Uncertainty gap $G$ and its rate of change $dG/dt$ on Phi-4-mini with QE method. The scatter points represent empirical data from self-improvement on the Math and GSM8k datasets, while the solid lines show the linear regression fits.

Equation (10) and Equation (11) demonstrate that uncertainties $U_s$, $U_v$, and their gap $G(t)$ all converge exponentially toward their respective limits. The proof of Proposition 3.1 is shown in Appendix B.1. We derive two key corollaries from Proposition 3.1.

**Corollary 3.1** *The partial derivative of the solver's final uncertainty with respect to the initial gap is a negative constant:* $\frac{\partial U_{s,\infty}}{\partial G_0} = -\frac{\beta}{\alpha-\beta}$.

The proof of Corollary 3.1 is shown in Appendix B.2. Corollary 3.1 shows that a larger initial verifier-solver gap ($G_0$) leads to a lower final solver uncertainty ($U_{s,\infty}$). This aligns with the core intuition that the performance gap is the driver of the self-improvement process.

**Corollary 3.2** *For a given tolerance $\epsilon > 0$, the training time $t$ required to ensure the Capability Gap is within $\epsilon$ of its convergence limit satisfies:* $t > \frac{\ln(\delta/\epsilon)}{k(\alpha-\beta)}$.

The proof of Corollary 3.2 is shown in Appendix B.3. Corollary 3.2 provides theoretical guidance for selecting the total number of training epochs $T$ in practical application. In practice, we first estimate the key convergence parameters by training for a few initial epochs and fitting the early-stage performance data. These parameters can then be used to determine an appropriate total training time $T$ to balance performance with computational costs.

The dynamics indicate that (i) the gap potential energy $E(t)$ drives the solver capability more strongly; (ii) the change of solver capability and verifier capability slows down as the gap decreases; and (iii) the capability gap might not converge to zero during the training process.

## 4 EXPERIMENT

This section conducts experiments on self-improvement to verify the theoretical framework. We first introduce experimental setups in Section 4.1. We then present experiment results of the self-improvement process in Section 4.2. We finally explore the differences between solver capability and verifier capability in Section 4.3, showing that the solver-verifier gap generally happens in practice.

### 4.1 SETUP

This section introduces the models, datasets, and key parameters. We consider the following two verification methods: TrueFalse (TF) and Quality Evaluation (QE). We utilize two model families: (a) **Phi models:** From the Phi family (Abdin et al., 2024), we use Phi-4-mini, Phi-3.5-mini, and Phi-3-mini; (b) **Llama models:** We use Llama-3.2-3B (Grattafiori et al., 2024) and Llama-3.1-8B. Our experiments mainly focus on the models' mathematical problem-solving capabilities. Accordingly, we employ two representative datasets: GSM8k (Cobbe et al., 2021) and Math (Hendrycks et al., 2021). In addition, we also consider ProntoQA and MBPP datasets with TF method in order to evaluate our framework on QA and code generation problems. Experimental details are provided in Appendix D.

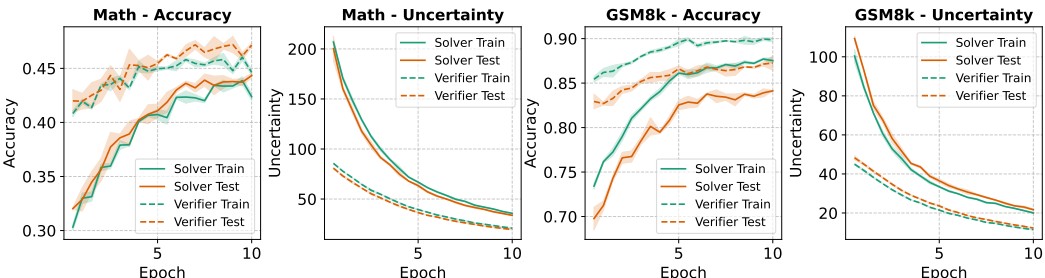

Figure 3: Accuracy and uncertainty during the self-improvement of the Phi-4-mini model on the Math and GSM8k datasets using the QE method. The experimental results show that the accuracy increases during self-improvement process while the uncertainty decreases.

## 4.2 Dynamics of Self-Improvement

This section sketches the dynamics of self-improvement using AI-generated feedback for supervised fine-tuning (SFT), aiming to verify the theoretical framework proposed in Section 3.2. We test the Phi-4-mini, Phi-3.5-mini, Phi-3-mini, and Llama-3.2-3B models. We apply Low-Rank Adaptation (LoRA) (Hu et al., 2022), which significantly reduces the number of updatable parameters, thereby enhancing SFT efficiency. The chosen hyperparameters are detailed in Appendix D.1.

**Results.** We present the results of self-improvement on the Phi-4-mini model using the QE method, as depicted in Figure 3. The empirical evidence indicates a consistent enhancement in the accuracy of both the solver and the verifier during the self-improvement process, coupled with a concurrent reduction in their respective uncertainties. Furthermore, a narrowing of the gap $G(t)$ between the solver and verifier is evident. Results for other models and methods are presented in Appendix D.2.

**Validation.** To validate our theoretical framework, we fit an exponential model to the uncertainty from 10 self-improvement epochs. Figure 4 presents the results for the Phi-4-mini model in four different settings (Math / GSM8K datasets train split with QE / TF metrics), which illustrates the evolution of three key metrics: solver uncertainty, verifier uncertainty, and the uncertainty gap. In Figure 4, the exponential model demonstrates a strong fit to the empirical data, with the coefficients of determination ($R^2$) exceeding 0.9. This empirical evidence validates the exponential law proposed in our theoretical work. Results for other models are presented in Appendix D.2.

## 4.3 Verifier Outperforms Solver

In this section, we evaluate the solver and verifier performance of LLM, aiming to validate the utility of the solver-verifier gap used in Section 3. Specifically, the experimental results verify that verifier capability outperforms solver capability consistently. Our evaluation is divided into two settings: self-evaluation and cross-evaluation. Self-evaluation employs the same model for both the solver and verifier roles, whereas cross-evaluation utilizes different models for the solver and verifier.

**Self Evaluation.** We compare the accuracy and uncertainty of solver and verifier on different models and datasets across different $N$. Given that the verifier selects one response from $N$ candidates generated by the solver, the verifier's performance is expected to improve as $N$ increases. Detailed results are presented in Appendix D.2.

**Cross Evaluation.** To better understand the relationship between solver and verifier capabilities, we also perform cross-evaluation, where one model acts as the solver and a different model acts as the verifier. Results are presented in Appendix D.2.

These experiments indicate that the verifier typically outperforms the solver, revealing a consistent positive performance gap between the verifier and the solver across model-task pairs. This performance gap is considered a key driver of self-improvement dynamics.

## 4.4 Impact of Training Stages on Self Improvement

In this section, we investigate how upstream training stages influence the self-improvement dynamics, we conducted a controlled comparative analysis. This section addresses the potential confounder that models at different stages might exhibit distinct optimization behaviors. We utilize the EvoLM model

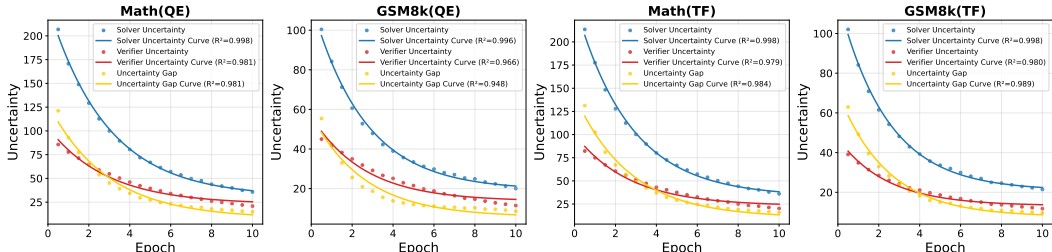

Figure 4: Exponential trends of model uncertainty during self-improvement on train split. The results illustrate the uncertainty and gap of the solver and verifier for Phi-4-mini. The scatter points represent the measured data, while the solid lines are the best-fit curves to an exponential model. $R^2 > 0.9$ indicates that the evolution of these uncertainties is well-described by an exponential function.

suite Qi et al. (2025) for this study, which allows us to control for model architecture (Llama-2 1B) while isolating the impact of training stages. We compared three checkpoints representing the full lifecycle:

- Standard SFT Baseline (Base Stage): Derived from the raw pre-trained model (1B-160BT) via standard SFT. This represents a model with high latent potential but no domain-specific mid-training.

- Mid-train Enhanced Model (Mid-train Stage): Derived from a model that underwent domain-specific Continued Pre-training (CPT) on math data (1B-160BT-8+42BT) before SFT. This represents the domain-primed state.

- Fully Post-trained Model (Post-train Stage): A model that has completed the full pipeline, including mid-training, SFT and Reinforcement Learning (1B-160BT-8+42BT-100Kep1-100Kep16). This represents the converged state.

We present the results in Figure 5 and Figure 9. The results shows a clear evolution in training dynamics governed by model training stage: Base model exhibit the highest plasticity, driven by the largest initial gap. Mid-training lowers the initial potential energy and increases initial capability. Post-trained models demonstrate dynamic saturation, validating the asymptotic convergence as the capability gap is minimized.

## 5 DISCUSSIONS ON CROSS-IMPROVEMENT

In this section, we discuss how to model cross-improvement within the framework described in Section 3.2. The key insight is that external data affects the theoretical framework through the verification capability. We first present the theoretical framework in Section 5.1, under the framework with limited external data. We then conduct experiments in Section 5.2 to validate the theoretical findings.

### 5.1 THEORETICAL FRAMEWORK OF CROSS-IMPROVEMENT

In this section, we present the theoretical framework of training dynamics of cross-improvement. We follow the notations in Section 3. Besides, assume that we have limited external data with size $M$. For example, we may acquire $M$ external data in total from a better LLM using API queries.

We focus on the allocation of the external data. Specifically, for each epoch $t$, only $\eta_t M$ prompts could use API queries to get (one) external data, with $\sum_{t=1}^{T} \eta_t = 1$ where $T$ denotes the total training epochs. For those chosen prompts, we choose the external data as the BoN response; for those non-chosen prompts, we still choose the BoN data from the $N$ internal responses. Notably, as long as one prompt is chosen to use external data, it will always use the external data.

**Cross-Data Effects.** In the cross-improvement framework, the use of external data will influence the verifier capability $U_v(t)$, since we use a different definition of BoN which directly relates to the verifier capability. To model the effects, we assume the verifier capability after cross-improvement as

$$U_v^c(t) = (1 + \gamma \eta_t)^{-1} U_v(t-1). \tag{12}$$

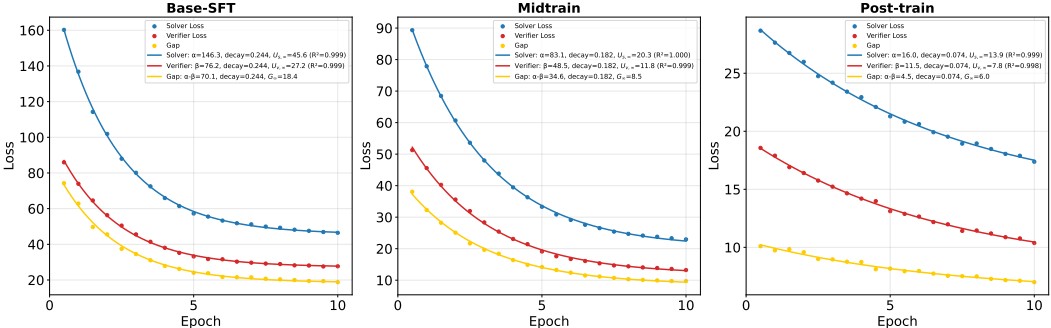

Figure 5: Self-improvement dynamics across Base, Mid-trained, and Post-trained stages on train split. Fitted curves ($R^2$) reveal a clear evolutionary trajectory: (1) The Base-SFT model exhibits the highest Solver Rate ($\alpha = 146.3$) and Decay Rate (0.244). (2) The Mid-trained model reduces the initial uncertainty, proving that mid-training optimizes the initialization state. (3) The Post-trained model shows the lowest rate ($\alpha = 16.0$), confirming the theory of saturation near the capability limit.

The parameter $\gamma$ represents the general effect of cross-improvement, while $\eta_t$ represents the ratio of external data used in epoch $t$. We assume that the effect $\gamma$ is time invariant, $\gamma > 0$ when the verifier that adds external data performs better than the verifier without external data; otherwise $\gamma < 0$. Overall, in each step, the cross-data first boosts the verification capability, then it evolves following the mathematical framework in Section 3.2. We illustrate this procedure in Figure 7.

**Solutions.** We model the above framework on cross-improvement as

$$U_s(t)|_{t=0} = U_{s,0}, \qquad U_v(t)|_{t=0} = U_{v,0}, \tag{13}$$

$$U_s^c(t) = U_s(t-1), \qquad U_v^c(t) = (1+\gamma\eta_t)^{-1}U_v(t-1), \tag{14}$$

$$G^c(t) = U_s^c(t) - U_v^c(t), \qquad E(t) = kG^c(t) - b, \tag{15}$$

$$U_s(t) - U_s^c(t) = -\alpha E(t), \quad U_v(t) - U_v^c(t) = -\beta E(t). \tag{16}$$

Equation (13) represents the initial conditions, which are the same as Equation (7). Equation (14) represents the effects of cross-improvement, where the solver capability after cross-improvement $U_s^c(t)$ remains unchanged, while the verifier capability increases as discussed in Equation (12). The current capability gap $G^c(t)$ is then defined as the gap between the solver capability and the current verifier capability. Equation (16) represents the iterative conditions of cross-improvement. Based on the above formulation, we derive an approximate solution of the ultimate uncertainty:

**Proposition 5.1** *Let the external-data-affected solver uncertainty $U_s^c(t)$ and verifier uncertainty $U_v^c(t)$. The ultimate uncertainty vector $\mathbf{U}(T)$ can be approximated as*

$$\mathbf{U}(T) \approx e^{-\mathbf{\Delta}'}\mathbf{U}(0), \quad \mathbf{\Delta}' = \begin{pmatrix} T - (1+\beta k)(T - \gamma\sum_{t=1}^T \eta_t) & T\beta k & -T\beta b \\ -\alpha k(T - \gamma\sum_{t=1}^T \eta_t) & T\alpha k & -T\alpha b \\ 0 & 0 & 0 \end{pmatrix}, \tag{17}$$

*where $\mathbf{U}(T)$ denotes $[U_{v,T}, \quad U_{s,T}, \quad 1]^\top$ and $\mathbf{U}(0)$ denotes $[U_{v,0}, \quad U_{s,0}, \quad 1]^\top$. $U_s(T)$ is related only to the summation $\sum_{t=1}^T \eta_t$ (instead of each $\eta_t$ at epoch $t$).*

Under Proposition 5.1, we come to the following conclusions: (i) For cross-improvement with $\sum_{t=1}^T \eta_t = 1$, the approximate solution is around the same; (ii) The cross-improvement with $\sum_{t=1}^T \eta_t = 1$ outperforms self-improvement with $\sum_{t=1}^T \eta_t = 0$ in terms of solver capability, when $\gamma > 0$. The proof of Proposition 5.1 is shown in Appendix B.4.

## 5.2 EXPERIMENTS

In this section, we perform cross-improvement experiments using different allocation strategies. Figure 7 illustrates the process of cross-improvement. For these experiments, external data are generated by DeepSeek-V3 which achieves accuracy of $60.47\%$ and $91.18\%$ on Math and GSM8k

Table 1: Solver accuracy with QE method on train split: raw data and relative improvements of strategies average (%). Initial represents the performance of the original model, while Baseline represents the results from self-improvement. We deploy three strategies: Early, Uniform, and Late. The last row shows the difference in verifier accuracy between using all external data at the start (Early strategy at $t = 0$) and the baseline's initial verifier accuracy (Initial).

| | Phi-4-mini | | Llama-3.2-3B | |
|---|---|---|---|---|
| Strategy | Math(%) | GSM8k(%) | Math(%) | GSM8k(%) |
| Initial | 30.31 ($\pm$ 0.24) | 73.42 ($\pm$ 0.33) | 36.02 ($\pm$ 0.25) | 63.10 ($\pm$ 0.58) |
| Baseline | 45.08 ($\pm$ 0.48) | 88.53 ($\pm$ 0.48) | 49.16 ($\pm$ 0.90) | 88.66 ($\pm$ 0.73) |
| Early | 46.33 ($\pm$ 0.16) | 88.54 ($\pm$ 0.12) | 53.48 ($\pm$ 0.41) | 88.26 ($\pm$ 0.57) |
| Uniform | 46.56 ($\pm$ 0.19) | 88.44 ($\pm$ 0.05) | 52.74 ($\pm$ 0.82) | 88.23 ($\pm$ 0.58) |
| Late | 45.83 ($\pm$ 0.48) | 88.90 ($\pm$ 0.21) | 51.31 ($\pm$ 0.76) | 87.19 ($\pm$ 0.28) |
| Max-Min | 0.73 | 0.46 | 2.17 | 1.07 |
| Avg | 46.24 | 88.63 | 52.51 | 87.89 |
| Avg vs Initial | +15.93 | +15.21 | +16.49 | +24.79 |
| Avg vs Baseline | +1.16 | +0.10 | +3.35 | -0.77 |
| Early($t = 0$) vs Initial (Verifier) | +5.87 | +0.96 | +0.97 | -1.58 |

dataset respectively. Specifically, we utilize DeepSeek-V3 responses for fine-tuning instead of BoN responses. We test three allocation strategies: (a) **Early:** All external data are introduced in the first epoch. (b) **Uniform:** An equal amount of new external data is introduced in each epoch. (c) **Late:** All external data are introduced in the eighth epoch.

We present the results on the training data in Table 1. We observe that the average solver accuracy for all three strategies is higher than the baseline on the MATH dataset. However, Phi-4-mini shows a slight performance improvement on GSM8k, while Llama-3.2-3B exhibits a drop on GSM8k dataset. This observation indicates that a key factor influencing the effectiveness of cross-improvement is indeed whether the external model's verifier capability significantly surpasses that of the original model. In addition, we find that the difference in accuracy between the three strategies is slight, which validates the conclusions under Proposition 5.1.

## 6 CONCLUSION

In this paper, we propose a theoretical framework to analyze the training dynamics of self-improvement via the solver-verifier gap. Experimental results on various datasets and models accord well with the theoretical findings. Besides, one may derive the theoretical limits based on the framework, which is closely related to the model's verification capability. To break the limit, one may apply cross-improvement to enhance the model's verification capability. Therefore, we further introduce the corresponding theoretical framework on the dynamics of cross-improvement under limited external data regimes, and find that the allocation of external data might have less influence on the final results. Experimental results verify the theoretical findings.

In the end, we provide several future directions: (i) We assume a time-invariant property when analyzing the cross-improvement, which might be relaxed in future work; (ii) External data can be used to fine-tune a self-improved model to further improve the performance of the model; (iii) We use a phenomenological modeling approach in our framework. A deeper investigation into the underlying mechanisms that cause the self-improvement dynamics to be well-described by our physics-inspired model would be a key future direction. This includes further research into the relationship between the model's key parameters, $\alpha$ and $\beta$, and factors such as the LLM's architecture.

## ETHICS STATEMENT

This research is theoretical and methodological, focusing on developing a mathematical model for the training dynamics of LLM self-improvement. All experiments were conducted on public academic benchmarks and did not involve human subjects or sensitive data, thus requiring no Institutional Review Board (IRB) approval. Our work aims to deepen the understanding of the self-improvement mechanism to foster the development of more predictable and reliable models.

## REPRODUCIBILITY STATEMENT

To ensure the reproducibility of our work, we provide detailed descriptions of our theoretical and experiment setup. Our theoretical model is detailed in Section 3, with complete mathematical proofs for all propositions and corollaries provided in Appendix B. The experiment setup is described in Section 4.1. Further implementation details, such as the self-improvement algorithm (Algorithm 1) and the hyperparameters used to obtain the results (Table 3), are available in Appendix D.

## ACKNOWLEDGMENTS

We would like to thank the members of the Jiaye's Group for their insightful discussions throughout this project. We are also grateful to the anonymous reviewers for their constructive comments which helped improve the paper. This work is supported by Shanghai Science and Technology Development Funds 24YF2711700 and Fundamental Research Funds for the Central Universities 2024110586. Xin Liu is partially supported by the National Natural Science Foundation of China 12201383 and Innovative Research Team of Shanghai University of Finance and Economics. Yifan Sun would like to express his deepest gratitude to his beloved girlfriend, Jinjin Zheng, for her unwavering companionship and support, which served as a constant source of strength throughout the inevitable lows of this research journey. Yushan Liang would like to extend her special thanks to her parents for their unwavering support.

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

# Appendix

This part provides the supplementary materials. Appendix A provides necessary details omitted before. Appendix B provides omitted derivations of the theoretical framework. We claim the use of LLMs in Appendix C. Appendix D provides experiment details and omitted experiment results.

## A  SUPPLEMENTARY DETAILS

In this section, we provide the details omitted before. Appendix A.1 presents the omitted algorithm and illustration, while Appendix A.2 summarizes main contributions of our work. In Appendix A.3, we discuss the relationship between uncertainty and capability. Additionally, we further discuss the difference between our work and prior works, as well as the challenges for self-improvement in Appendix A.4.

### A.1  OMITTED ILLUSTRATION

Figure 6 gives an overview of our theoretical framework. Algorithm 1 presents one-step self-improvement used in our experiments while Figure 7 illustrates cross-improvement process.

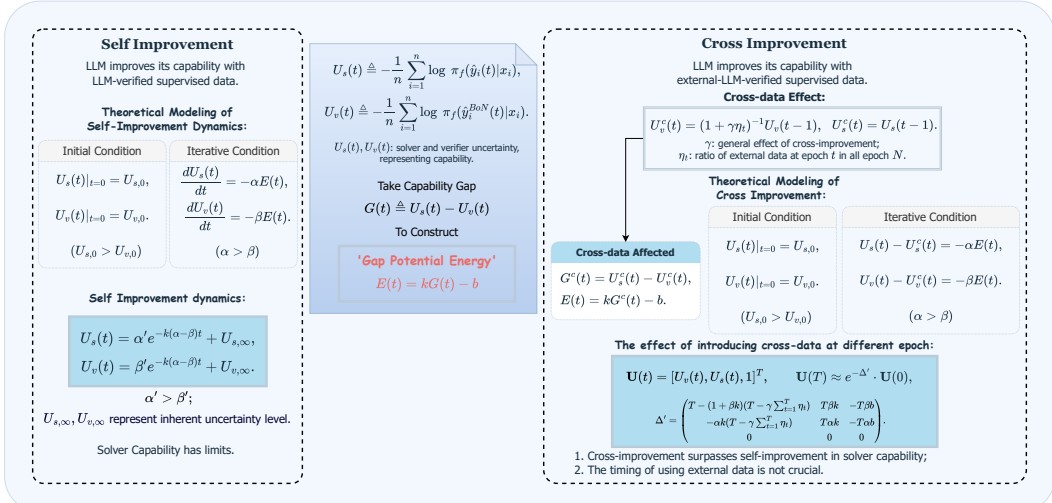

Figure 6: Overview of the theoretical framework on self-improvement and cross-improvement.

### A.2  CONTRIBUTIONS

In this section, we summarize main contributions of our work. Our main contributions are:

- **Theoretical framework:** We detail our theoretical framework on the dynamics of self-improvement in Section 3, inspired by the potential energy framework based on the *solver-verifier capability gap* in Equation (6). Theoretical derivations in Equation (11) imply an exponential law for the model capability, and we further establish the extreme capability based on the framework.
- **Experiments:** We conduct experiments on self-improvement to verify the theoretical framework in Section 4 across multiple models, datasets, and verification methods. Our empirical observations indicate that (i) the uncertainty/accuracy during the self-improvement process follows an exponential law with an extreme capability (Figure 3) as implied by the theoretical framework; (ii) the solver-verifier gap generally happens in practice across different regimes (Section 4.3), validating the utility of the gap as the potential energy.
- **Cross-improvement:** We further investigate in Section 5 the cross-improvement under the above framework, where limited external data is provided during the training process. We contend in Section 5.1 that cross-improvement might improve the verifier capability, thus improving the extreme capability of self-improvement. Empirically, we observe in Section 5.2 that if the verifier

---

**Algorithm 1** Self Improvement (One Step)

---

**Require:** Prompts set $\mathcal{X}$:$\{x_1, x_2, \cdots, x_n\}$, Current model $f$, Sample Size $N$, Threshold $\sigma$
1: $\mathcal{Y}^{\text{BoN}} \leftarrow \emptyset, \mathcal{U}^{\text{BoN}} \leftarrow \emptyset$;
2: **for** each prompt $x_i \in \mathcal{X}$ **do**
3:    $C_{x_i} \leftarrow \emptyset$;
4:    **for** $j \leftarrow 1$ to $N$ **do**
5:       Generate response $\hat{y}_{i,j} \sim \pi_f(\cdot|x_i)$;
6:       Ask model to evaluate $\hat{y}_{i,j}$ and return a score $s(\hat{y}_{i,j})$;
7:       **if** $s(\hat{y}_{i,j}) \geq \sigma$ **then**
8:          Append $\hat{y}_{i,j}$ to $C_{x_i}$;
9:       **end if**
10:    **end for**
11:    $\hat{y}_i^{\text{BoN}} \leftarrow \arg\min_{C_{x_i}} \frac{1}{L(\hat{y}_{i,j})} U_f(\hat{y}_{i,j}|x_i)$;
12:    Append $\hat{y}_i^{\text{BoN}}$ to $\mathcal{Y}^{\text{BoN}}$;
13:    Append $U_f(\hat{y}_i^{\text{BoN}})$ to $\mathcal{U}^{\text{BoN}}$;
14: **end for**
15: Uncertainty $\leftarrow \frac{1}{|\mathcal{U}^{\text{BoN}}|} \sum_{u \in \mathcal{U}^{\text{BoN}}} u$;
16: $\hat{f} \leftarrow \text{AdamW}(f, \text{Uncertainty})$;
**Ensure:** Self-improved model $\hat{f}$

---

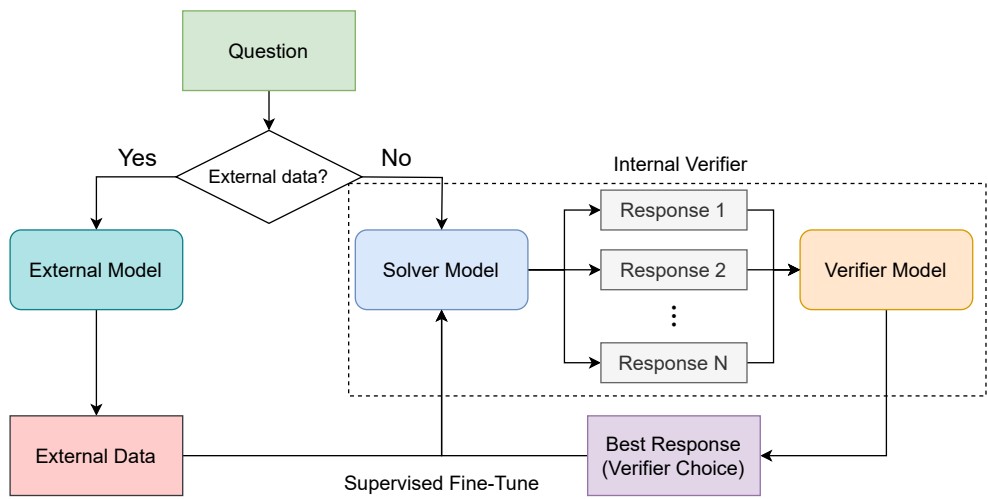

Figure 7: A diagram of Cross Improvement

capability is improved by the external data, the performances of LLM are enhanced, and vice versa (Table 1). This accords with the theoretical findings.

## A.3 RELATIONSHIP BETWEEN UNCERTAINTY AND CAPABILITY

In this section, we clarify the relationship between uncertainty and capability. Although low uncertainty is often correlated with high capability (i.e., accuracy), they are not conceptually equivalent. In this work, we use uncertainty instead of accuracy as a proxy metric for model capability. The justifications are detailed as follows:

- **Sharpening Mechanism:** Following Huang et al. (2025), minimizing the loss ($U_v$) forces the model to sharpen its intrinsic probability distribution onto high-quality responses. A decrease in $U_s$ signifies that this distributional shift has occurred and that the model has successfully learned from the verifier's output. Thus, $U_s$ is the direct measure of the model's capability.

- **Empirically correlation between accuracy and uncertainty:** To validate that $U_s$ indicates the model's capability, we calculate the Pearson correlation coefficient (r) between $U_s$ and solver accuracy for different models and datasets (Table 2). Accuracy and Uncertainty for most model-dataset pairs are significantly negatively correlated ($|r| > 0.8$), which proves that reducing solver uncertainty ($U_s$) translates to gains in accuracy.

- **Absence of Ground Truth:** In many real-world scenarios, particularly open-ended generation tasks, ground-truth labels are often unavailable or difficult to define. Unlike accuracy, which strictly depends on external gold references, uncertainty is computed from the model's output distribution. This makes uncertainty a necessary and universal metric for monitoring capability evolution in label-free regimes where accuracy calculation is infeasible.

- **The Optimization Objective:** In standard SFT, the objective of training is to maximize the likelihood of correct responses, which is mathematically equivalent to minimizing their uncertainty (Negative Log-Likelihood). In our framework, we regard the output of the verifier as the ground truth for SFT. The loss function at step $t$ is defined as $L_t(f) \triangleq U_v(t) = -\frac{1}{n} \sum_{i=1}^{n} \log \pi_f(\hat{y}_i^{\text{BoN}}(t)|x_i)$, which is identical to our definition of verifier uncertainty $U_v(t)$.

Table 2: Correlation coefficients between Uncertainty and Accuracy.

| Dataset | Method | Llama-3.2-3B | Phi-3-mini | Phi-3.5-mini | Phi-4-mini |
|---------|--------|--------------|------------|--------------|------------|
| GSM8k | QE | $-0.963^{***}$ | $-0.985^{***}$ | $-0.974^{***}$ | $-0.991^{***}$ |
|  | TF | $-0.966^{***}$ | $-0.990^{***}$ | $-0.967^{***}$ | $-0.988^{***}$ |
| Math | QE | $-0.965^{***}$ | $-0.958^{***}$ | $0.177$ | $-0.983^{***}$ |
|  | TF | $-0.967^{***}$ | $-0.900^{***}$ | $0.167$ | $-0.978^{***}$ |
| ProntoQA | TF | $-0.867^{***}$ | $-0.897^{***}$ | $-0.764^{***}$ | $-0.850^{***}$ |
| MBPP | TF | $-0.865^{***}$ | $-0.951^{***}$ | $-0.936^{***}$ | $-0.6735^{*}$ |

Note: $^{***} p < 0.001$, $^{**} p < 0.01$, $^{*} p < 0.05$.

## A.4 Review and Prospect

In this section, we discuss the difference between our work and prior works. We also discuss the challenge and perspective of self-improvement.

**Different definition:** In previous work (Song et al., 2025), the solver capability is defined as the accuracy, while the verifier capability is defined as the accuracy of responses that the model deems good responses. The calculation of accuracy depends on the golden answer contained in the original dataset, which is the most significant difference from our definition. In practice, many datasets do not contain gold answers, so our definition could be more widely applied.

**Different concerns:** Most of the prior works focus on finding a method that could make self-improvement more efficient. Instead, we pay attention to model the dynamics of self-improvement, which helps to understand the progress of self-improvement. In addition, we propose a new theoretical framework for cross-improvement which could be regarded as a solution to break through the limit of self-improvement.

**Challenge:** (i) Under supervised fine-tuning, self-improvement may be misled by incorrect responses. Thus, ensuring that the verifier outputs correct responses is a challenge. (ii) The current self-improvement method can only improve the model's capability in similar tasks by optimizing a certain task. Finding a method to improve the performance of the model on different tasks is a challenge.

## B Theoretical Derivation

In this section, we present the detailed theoretical derivations omitted before.

## B.1 PROOF OF PROPOSITION 3.1

In this section, we present the detailed derivation of Proposition 3.1. We first derive Equation (10). From Equation (8) and linear approximation $f(G) \approx kG - b$, we have

$$\frac{dG(t)}{dt} \approx -(\alpha - \beta)(kG - b). \tag{18}$$

The general solution of Equation (18) is

$$G(t) \approx Ce^{-k(\alpha-\beta)t} + \frac{b}{k}, \tag{19}$$

where C is a constant. To find the constant $C$, we use the initial condition at $t = 0$: $G(0) = U_{s,0} - U_{v,0} = G_0$, thus

$$G(0) = Ce^0 + \frac{b}{k}, \tag{20}$$

$$C = U_{v,0} - U_{s,0} - \frac{b}{k}. \tag{21}$$

Let $\delta = U_{v,0} - U_{s,0} - \frac{b}{k}$, we have

$$G(t) \approx \delta e^{-k(\alpha-\beta)t} + \frac{b}{k}. \tag{22}$$

Then we derive Equation (11). From Equation (22), we have

$$E(t) \approx kG(t) - b = k(\delta e^{-k(\alpha-\beta)t} + b/k) - b = k\delta e^{-k(\alpha-\beta)t}. \tag{23}$$

Solving the original differential equation for $U_s(t)$:

$$\frac{dU_s(t)}{dt} = -\alpha E(t) \approx -\alpha(k\delta e^{-k(\alpha-\beta)t}), \tag{24}$$

$$\int dU_s(t) = \int -\alpha k\delta e^{-k(\alpha-\beta)t}dt, \tag{25}$$

$$\int -\alpha k\delta e^{-k(\alpha-\beta)t}dt = -\alpha k\delta \left(\frac{e^{-k(\alpha-\beta)t}}{-k(\alpha-\beta)}\right) + C_s, \tag{26}$$

$$U_s(t) \approx \frac{\alpha\delta}{\alpha-\beta}e^{-k(\alpha-\beta)t} + C_s. \tag{27}$$

Using the initial condition $U_s(0) = U_{s,0}$ to solve for the constant $C_s$:

$$U_{s,0} = \frac{\alpha\delta}{\alpha-\beta}e^0 + C_s \implies C_s = U_{s,0} - \frac{\alpha\delta}{\alpha-\beta}, \tag{28}$$

Let $\alpha' = \frac{\alpha\delta}{\alpha-\beta}$ and $U_{s,\infty} = U_{s,0} - \alpha'$. Thus, the solution for $U_s(t)$ is

$$U_s(t) \approx \alpha'e^{-k(\alpha-\beta)t} + U_{s,\infty}. \tag{29}$$

The derivation for $U_v(t)$ is analogous, yielding

$$U_v(t) \approx \beta'e^{-k(\alpha-\beta)t} + U_{v,\infty}, \tag{30}$$

where $\beta' = \frac{\beta\delta}{\alpha-\beta}$ and $U_{v,\infty} = U_{v,0} - \beta'$.

## B.2 PROOF OF COROLLARY 3.1

In this section, we present the detailed derivation of Corollary 3.1. From Proposition 3.1, the final solver uncertainty is given by $U_{s,\infty} = U_{s,0} - \alpha'$, where $\alpha' = \frac{\alpha}{\alpha-\beta}\delta$ and $\delta = G_0 - b/k$. Substituting these into the expression for $U_{s,\infty}$ yields $U_{s,\infty} = U_{s,0} - \frac{\alpha}{\alpha-\beta}(G_0 - \frac{b}{k})$. Assuming $U_{s,0}$ is constant, we have $\frac{\partial U_{s,0}}{\partial G_0} = 0$. Thus, we have $\frac{\partial U_{s,\infty}}{\partial G_0} = -\frac{\alpha}{\alpha-\beta}$.

### B.3 PROOF OF COROLLARY 3.2

In this section, we present the detailed derivation of Corollary 3.2. The objective of Corollary 3.2 is to find the time $t$ such that the capability gap $G(t)$ is within a tolerance $\epsilon$ of its convergence value $G_\infty$, which is expressed as:

$$|G(t) - G_\infty| < \epsilon. \tag{31}$$

Starting from the expression for $G(t)$ given in Proposition 3.1, we have:

$$G(t) \approx \delta e^{-k(\alpha-\beta)t} + G_\infty, \tag{32}$$

$$G(t) - G_\infty \approx \delta e^{-k(\alpha-\beta)t}. \tag{33}$$

Since $\delta$ is defined on the basis of the initial gap, we can assume $\delta > 0$. Thus, we have

$$\delta e^{-k(\alpha-\beta)t} < \epsilon, \tag{34}$$

$$t > \frac{\ln(\delta/\epsilon)}{k(\alpha-\beta)}. \tag{35}$$

### B.4 PROOF OF PROPOSITION 5.1

In this section, we present the theoretical derivation of Proposition 5.1. According to Section 5.1, the dynamics conditions are formulated as:

$$
\begin{aligned}
&U_s(t)|_{t=0} = U_{s,0}, && U_v(t)|_{t=0} = U_{v,0}, && \text{(36)}\\
&U_s^c(t) = U_s(t-1), && U_v^c(t) = (1+\gamma\eta_t)^{-1}U_v(t-1), && \text{(37)}\\
&G^c(t) = U_s^c(t) - U_v^c(t), && E(t) = kG^c(t) - b, && \text{(38)}\\
&U_s(t) - U_s^c(t) = -\alpha E(t), && U_v(t) - U_v^c(t) = -\beta E(t). && \text{(39)}
\end{aligned}
$$

Denote $\mathbf{U}(t)$ as vector $[U_v(t), \quad U_s(t), \quad 1]^\top$, $\mathbf{U^c}(t)$ as vector $[U_v^c(t), \quad U_s^c(t), \quad 1]^\top$, with the following relationship holding true:

$$\mathbf{U^c}(t) = \begin{pmatrix} \frac{1}{1+\gamma\eta_t} & 0 & 0 \\ 0 & 1 & 0 \\ 0 & 0 & 1 \end{pmatrix} \mathbf{U}(t-1). \tag{40}$$

With these notations, we can derive the following iteration from Equation (37) to (39):

$$\mathbf{U}(t) = (I - \mathbf{\Delta}_t) \cdot \mathbf{U}(t-1), \tag{41}$$

$$\mathbf{\Delta}_t = \begin{pmatrix} 1 - \frac{1+\beta k}{1+\gamma\eta_t} & \beta k & -\beta b \\ -\frac{\alpha k}{1+\gamma\eta_t} & \alpha k & -\alpha b \\ 0 & 0 & 0 \end{pmatrix}. \tag{42}$$

Based on the iteration, $\mathbf{U}(t)$ at the end of the $T$ epochs is then:

$$\mathbf{U}(T) = \prod_{t=1}^{T}(I - \mathbf{\Delta}_t) \cdot \mathbf{U}(0), \tag{43}$$

where $\mathbf{U}(0)$ denotes $[U_{v,0}, \quad U_{s,0}, \quad 1]^\top$. The meaning of $U_{v,0}, U_{s,0}$ is defined in Equation (39). Under the circumstances, the following approximation holds true:

$$\prod_{t=1}^{T}(I - \mathbf{\Delta}_t) \approx \prod_{t=1}^{T} e^{-\mathbf{\Delta}_t} = e^{-\sum_{t=1}^{T}\mathbf{\Delta}_t} = e^{-\mathbf{\Delta}'}, \tag{44}$$

$$\mathbf{\Delta}' = \begin{pmatrix} T - (1+\beta k)(T - \gamma\sum_{t=1}^{T}\eta_t) & T\beta k & -T\beta b \\ -\alpha k(T - \gamma\sum_{t=1}^{T}\eta_t) & T\alpha k & -T\alpha b \\ 0 & 0 & 0 \end{pmatrix}. \tag{45}$$

The first approximation is based on matrix Taylor expansion $e^{\mathbf{A}} = \sum_{k=0}^{\infty} \frac{\mathbf{A}^k}{k!}, \|\mathbf{A}\| < 1$, given matrix $\mathbf{\Delta}_t(t = 1, \dots, T)$ is relatively small.

The second approximation is based on the fact that $\eta_t(t = 1, \ldots, T)$ is a relatively small quantity, considering that the total epoch $T$ is a large number. Therefore, the difference matrix between $\boldsymbol{\Delta}_i$ and $\boldsymbol{\Delta}_j(i \neq j, i, j = 1, \ldots, T)$ will be close to zero, making any two matrices $\boldsymbol{\Delta}_i, \boldsymbol{\Delta}_j(i \neq j, i, j = 1, \ldots, T)$ approximately commutative.

The third approximation is based on the approximation $\frac{1}{1+\gamma\eta_t} \approx 1 - \gamma\eta_t$, given $\eta_t$ a small quantity. Equation (44) and (45) indicates that one can derive a solution of $U_s(T)$, which is only related to $\sum_{t=1}^{T} \eta_t$. Therefore, one can come to the conclusion that (i) solver uncertainty at the final epoch $T$ is approximately the same for cross-improvement with $\sum_{t=1}^{T} \eta_t = 1$, and (ii) with the two elements in matrix $-\boldsymbol{\Delta}'$ that $\gamma$ appears in both negatively correlated to the term $\gamma \sum_{t=1}^{T} \eta_t$, the final solver uncertainty is also negatively correlated to this term. This indicates that cross-improvement with $\gamma > 0, \sum_{t=1}^{T} \eta_t = 1$ outperforms self-improvement with $\gamma > 0, \sum_{t=1}^{T} \eta_t = 0$, in terms of solver capability.

## C  THE USE OF LARGE LANGUAGE MODELS

We utilize Large Language Models (LLMs) to enhance the clarity, conciseness, and grammatical accuracy of the paper. The model's suggestions help refine sentence structure and improve the overall readability of the paper.

## D  EXPERIMENTAL DETAILS

In this section, we detail our experiment hyperparameters and present all results of our experiment. In Appendix D.1, we detail the hyperparameters chosen in our experiments. We provide all experiment results in Appendix D.2. All of our experiments are run on 80G NVIDIA A800 GPUs.

### D.1  HYPERPARAMETERS

In this section, we detail the hyperparameters in Table 3.

Table 3: Hyperparameters for SFT

| Learning Rate | Weight Decay | Solver Temperature | Verifier Temperature |
|---|---|---|---|
| 1e-5 | 0.01 | 1 | 0.1 |
| Sample Size (N) | Max New Tokens | LoRA Rank | LoRA dropout |
| 16 | 512 | 16 | 0.5 |

### D.2  OMITTED FIGURES

In this section, we will provide experiment details including setup and figures omitted in Section 4.

**Verification Methods.** We use two verification methods TrueFalse and Quality Evaluation in the experiment. TrueFalse (TF): The solver generates $N$ responses, denoted $\hat{y}_{i,1}, \cdots, \hat{y}_{i,N}$, for a prompt $x_i$. The verifier is then tasked with answering whether each response $\hat{y}_{i,j}$ is correct. If the verifier deems a response $\hat{y}_{i,j}$ correct, its score $s(\hat{y}_{i,j})$ is set to 1; otherwise, it is set to 0; Quality Evaluation (QE): The solver generates $N$ responses, $\hat{y}_{i,1}, \cdots, \hat{y}_{i,N}$, for a prompt $x_i$. The verifier then assigns a continuous score $s(\hat{y}_{i,j})$ between 0 and 1 to each response based on its quality. A score of $s(\hat{y}_{i,j}) = 0$ indicates a completely incorrect answer, while $s(\hat{y}_{i,j}) = 1$ indicates a completely correct answer.

**Self Improvement.** In this part, we display figures omitted in Section 4. In self-improvement, we employed mini-batch gradient descent with a batch size of 256. For each model-task pair, we conducted training for 10 epochs, saving a checkpoint after processing half of the training data. We test the solver accuracy and the verifier accuracy, defined as the accuracy of the response and the BoN response, respectively. Figure 10 illustrates the results of self-improvement on Phi-3.5-mini, Phi-3-mini and Llama-3.2-3B model with QE method. We observe that most of model-task pairs have similar results with the Phi-4-mini model except for several pairs. We also present the results with TF

method in Figure 11. When self-improving the Phi-3.5-mini and Phi-3-mini models on Math data set, we observe that accuracy and uncertainty decrease simultaneously after several training epochs. The reason for this phenomenon might be LLM misleading by incorrect responses, as the BoN response may correspond to a incorrect answer with low uncertainty. The solution of this problem could be a future direction. Additionally, we note that the verifier accuracy of Llama-3.2-3B on GSM8k decreases as training progresses and is lower than the solver accuracy after 8 epochs of training. One possible reason for this phenomenon is that we use length-regularized log-likelihood to obtain the BoN responses, so the BoN responses tend to be longer responses. Long responses are more likely to contain repeated content, which may reduce accuracy. In addition, we present the self-improvement results on ProntoQA dataset with TF method Figure 12.

**Validation.** In this part, we present the validation results of Phi-3.5-mini, Phi-3-mini and Llama-3.2-3B on train split in Figure 14 and observe that the empirical data are strongly fitted to the exponential model for all models. Results of Phi-4-mini, Phi-3.5-mini, Phi-3-mini and Llama-3.2-3B on test split are presented in Figure 15. Experiment results on ProntoQA dataset are presented in Figure 16.

**Self Evaluation.** This part we provide the results of self-evaluation for different sample size $N$. For each model, we evaluate the accuracy and uncertainty of both the solver and the verifier, varying $N$ across the values $2, 4, 8, 16, 32, 64$. Figure 18 illustrates the accuracy and uncertainty of Phi-4-mini, Phi-3.5-mini, Phi-3-mini, Llama-3.2-3B and Llama-3.1-8B on Math and GSM8k with different sample size using TF and QE respectively, which shows that the verifier outperforms solver in all model-task pairs.

**Cross Evaluation.** In cross evaluation we set $N = 16$ because, as demonstrated in Figure 18, the accuracy improve slightly for $N > 16$. The results of QE method are presented in Figure 21 while results of TF method are presented in Figure 22. We observe that when a fixed model serves as the solver, the verifier's performance generally surpasses that of the solver, even when a different model is employed as the verifier.

**Pass@K.** We investigate the underlying reason for the limit of self-improvement, focusing on how a decrease in the model's response diversity leads to a diminishing potential energy, thereby causing the model's capability to plateau. Although the self-improvement paradigm shows great improvement in model capability, it iteratively leads to a performance plateau. To investigate the underlying cause of this saturation, we conduct the Pass@K experiment on the initial model and the self-improved model. Pass@K is a metric that calculates the proportion of prompts where at least one of the $K$ responses is correct. When $K$ is large, Pass@K could be used to measure the diversity of the solver. We present the result in Figure 23. We observe that when $K$ is small, Pass@K increases with the number of epochs of self-improvement, validating the self-improvement process. However, when $K$ is large, we observe a slight decrease in Pass@K, indicating that the diversity of the solver is reduced through self-improvement. The degradation in diversity is caused by the convergence to a certain response, which is a potential reason for the limit of self-improvement. Lack of ProntoQA dataset in experiment as the Pass@K of the QA problem is close to 1 when $K$ is large.

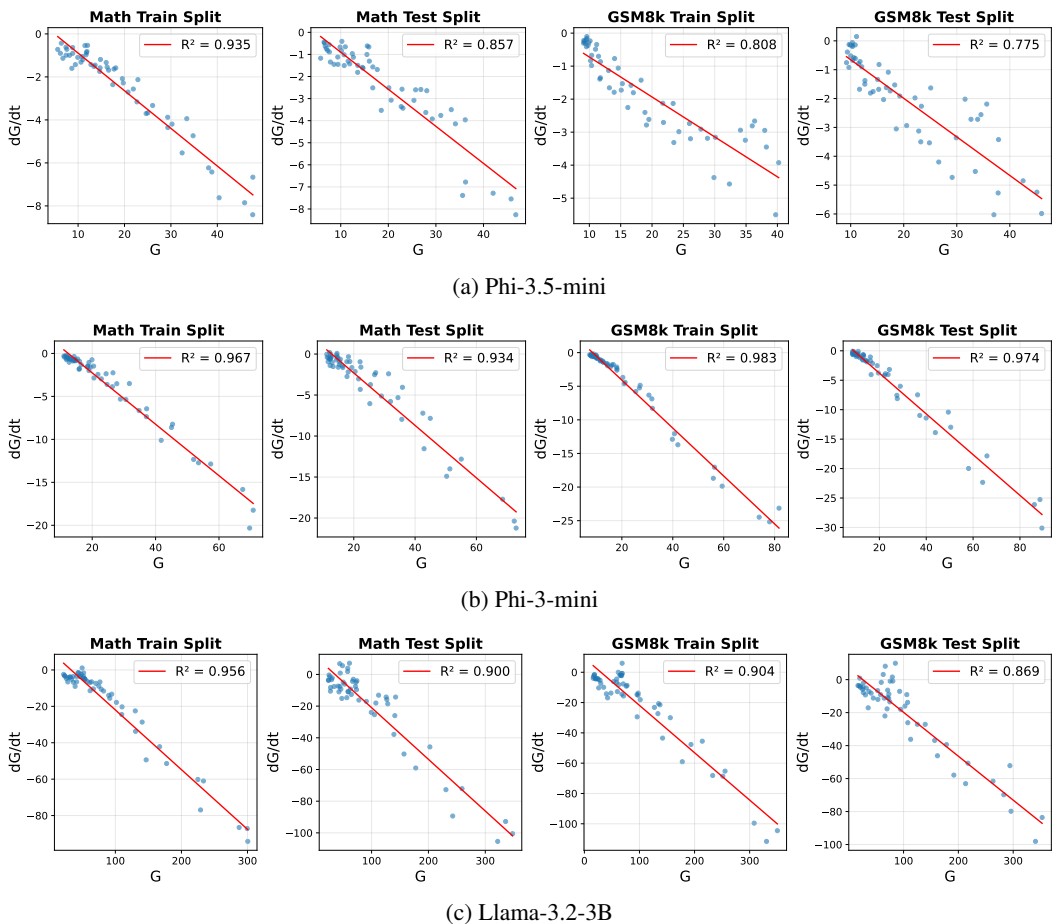

(a) Phi-3.5-mini

(b) Phi-3-mini

(c) Llama-3.2-3B

Figure 8: Validation of the linear relationship between the Uncertainty gap $G$ and its rate of change $dG/dt$ on Phi-3.5-mini, Phi-3-mini and Llama-3.2-3B with QE method. The scatter points represent empirical data from self-improvement on the Math and GSM8k datasets, while the solid lines show the linear regression fits.

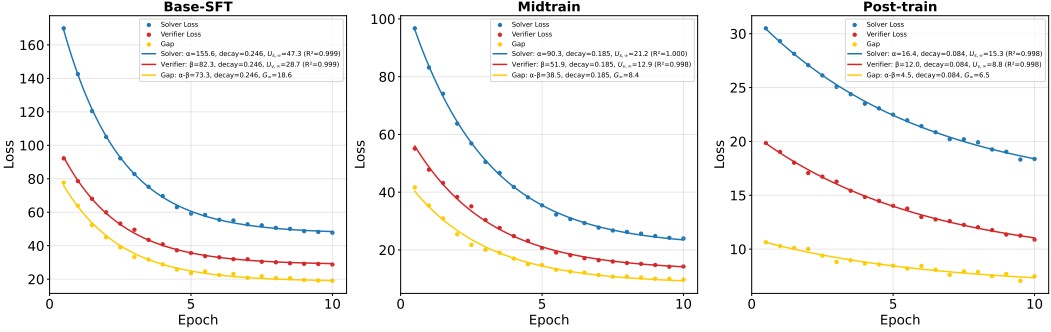

Figure 9: Self-improvement dynamics across Base, Mid-trained, and Post-trained stages on test split.

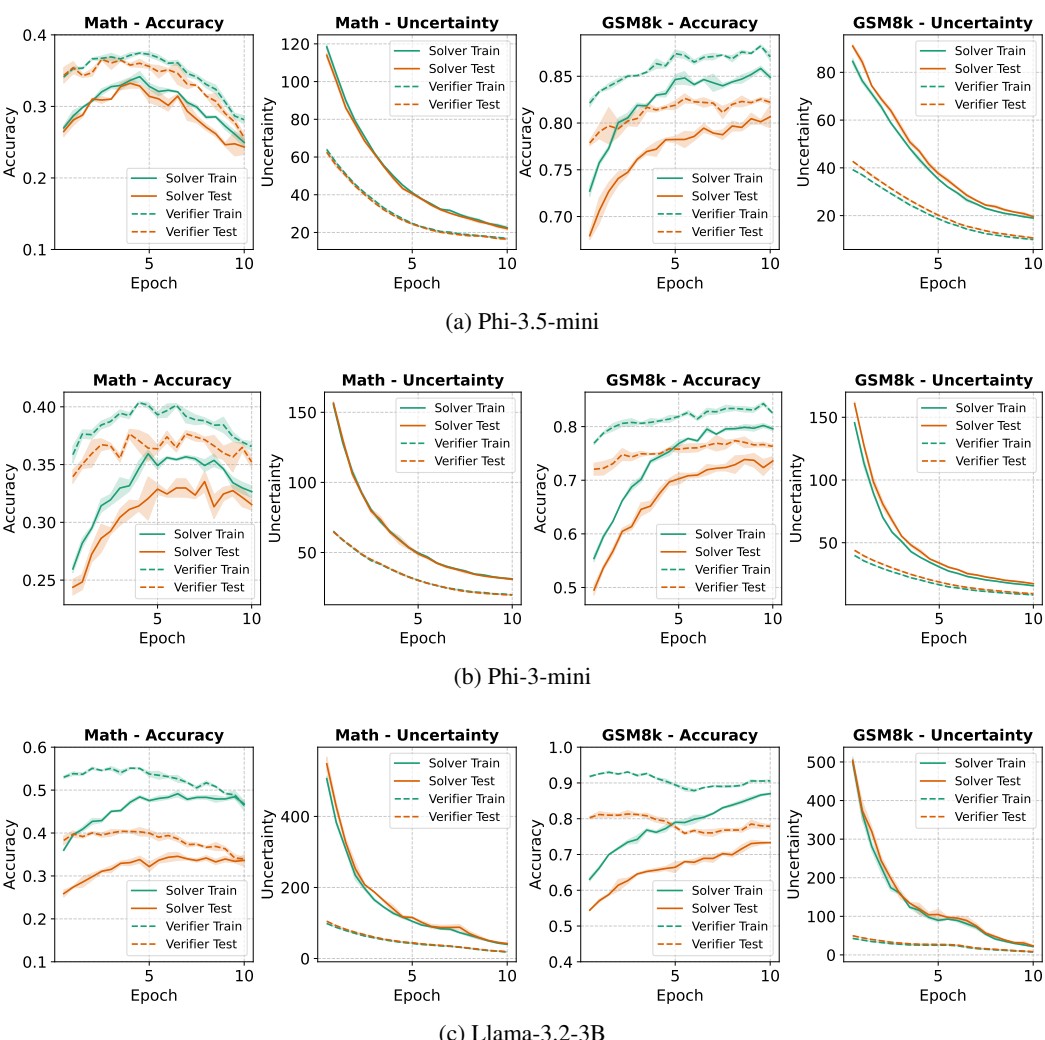

(a) Phi-3.5-mini

(b) Phi-3-mini

(c) Llama-3.2-3B

Figure 10: Accuracy and uncertainty during the self-improvement of the Phi-3.5-mini, Phi-3-mini and Llama-3.2-3B on the Math and GSM8k datasets using the QE method.

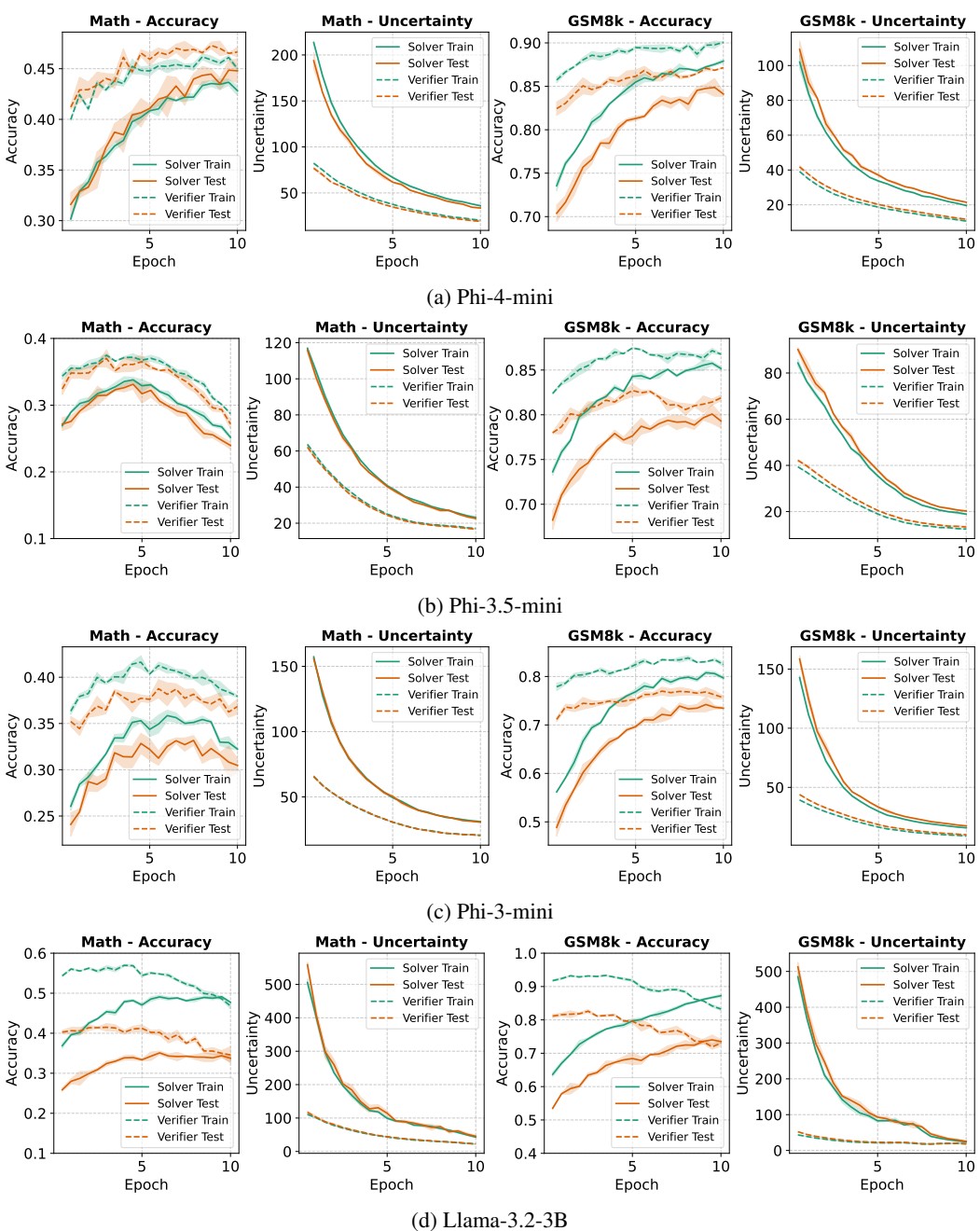

Figure 11: Accuracy and uncertainty during the self-improvement of the Phi-4-mini, Phi-3.5-mini, Phi-3-mini and Llama-3.2-3B on the Math and GSM8k datasets using the TF method.

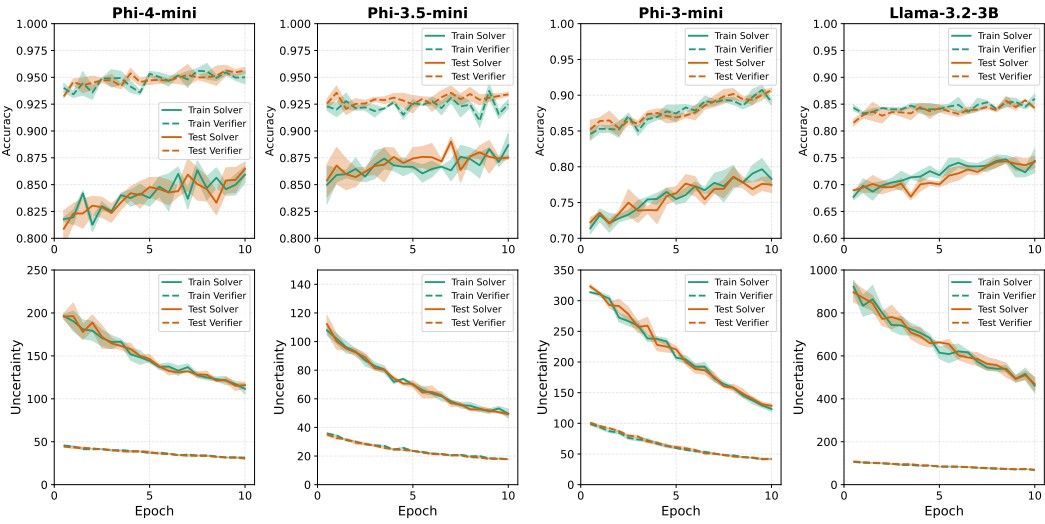

Figure 12: Accuracy and uncertainty during the self-improvement of the Phi-4-mini, Phi-3.5-mini, Phi-3-mini and Llama-3.2-3B on ProntoQA datset using the TF method.

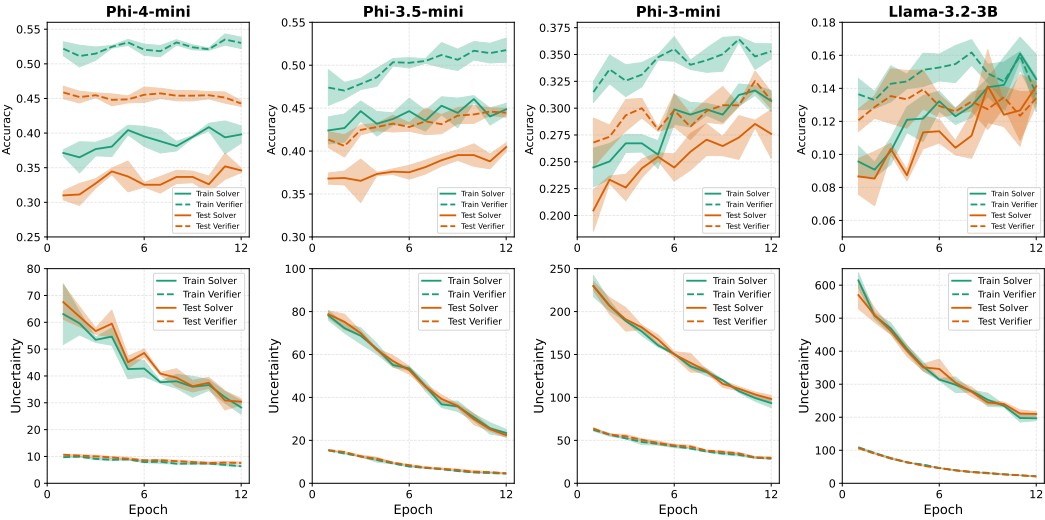

Figure 13: Accuracy and uncertainty during the self-improvement of the Phi-4-mini, Phi-3.5-mini, Phi-3-mini and Llama-3.2-3B on MBPP datset using the TF method.

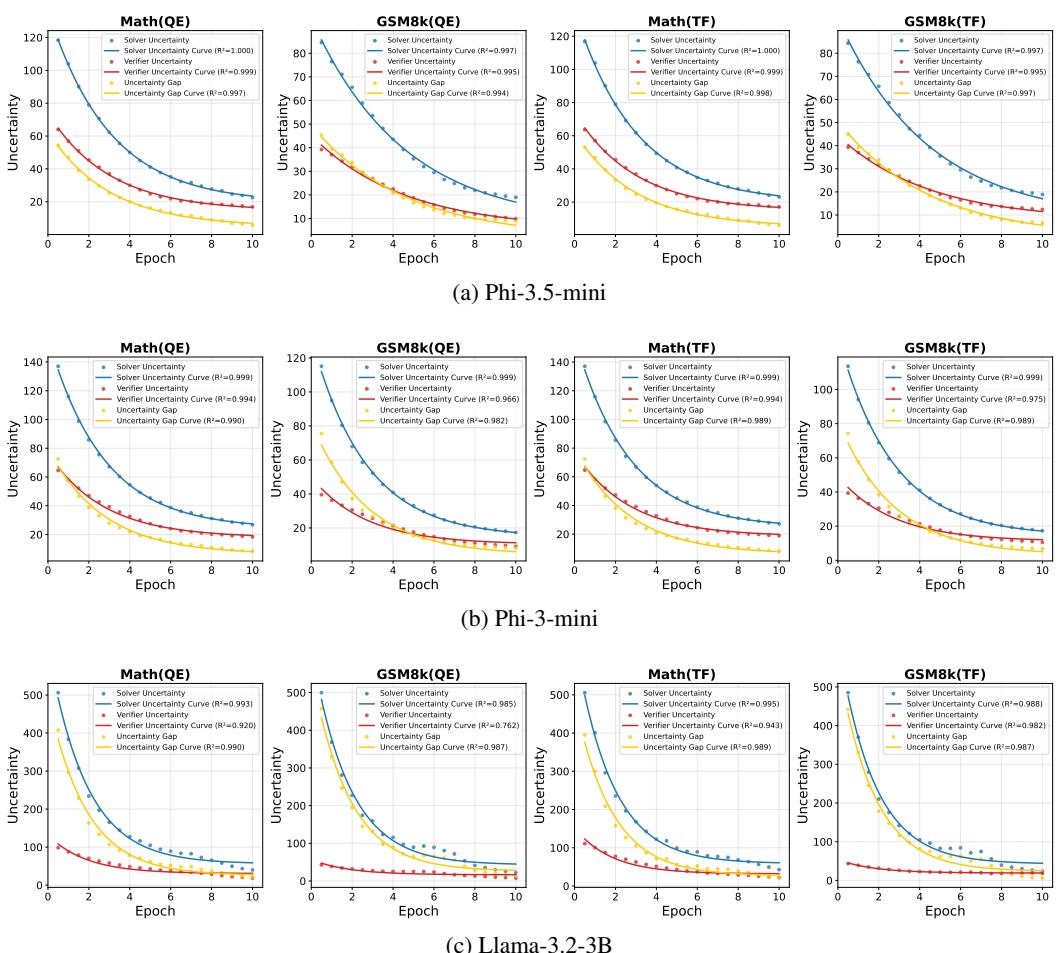

Figure 14: Uncertainty curve versus epochs of Phi-3.5-mini, Phi-3-mini and Llama-3.2-3B models using QE and TF methods on Math and GSM8k datasets train split.

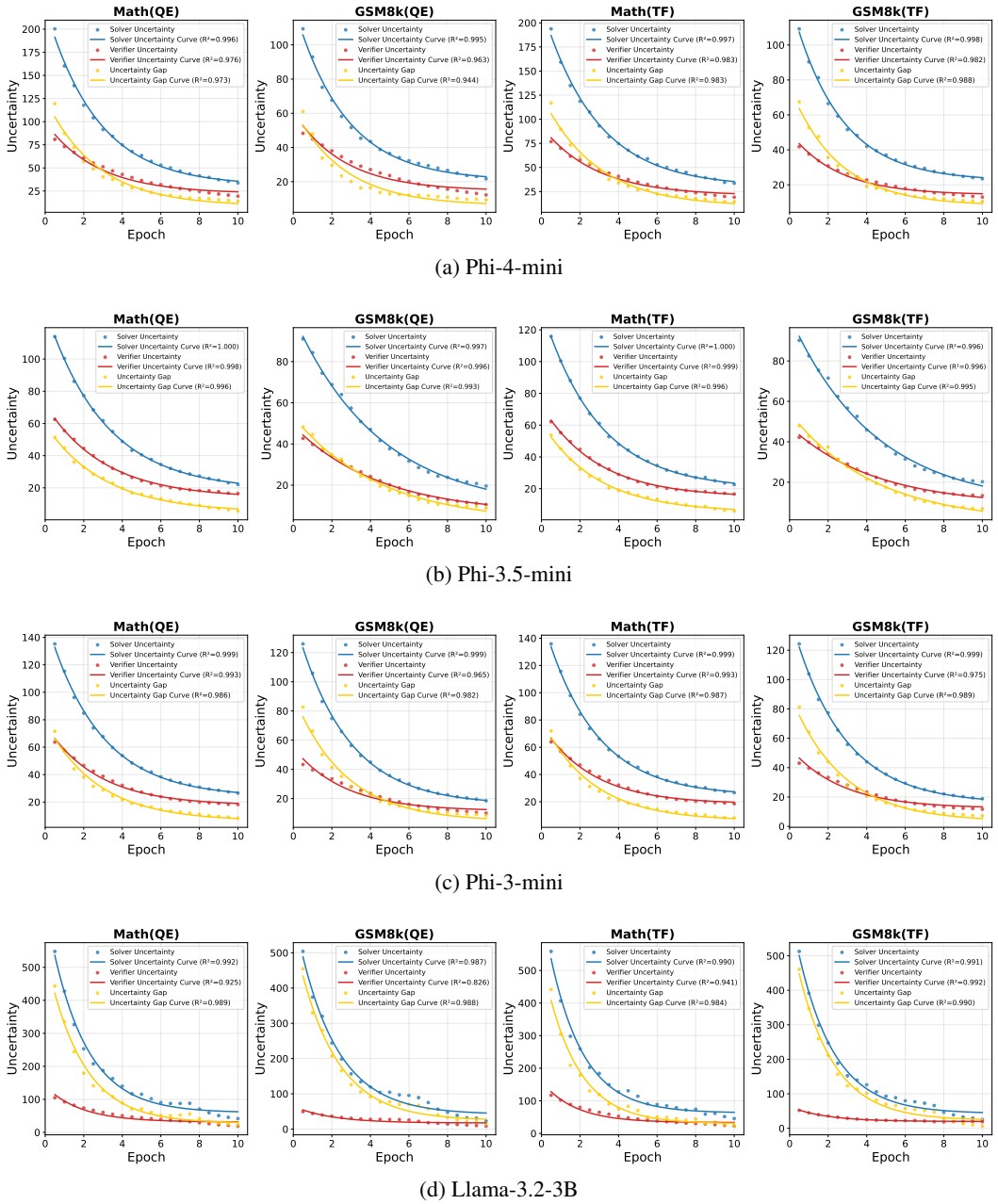

Figure 15: Uncertainty curve versus epochs of Phi-4-mini, Phi-3.5-mini, Phi-3-mini and Llama-3.2-3B models using QE and TF methods on Math and GSM8k datasets test split.

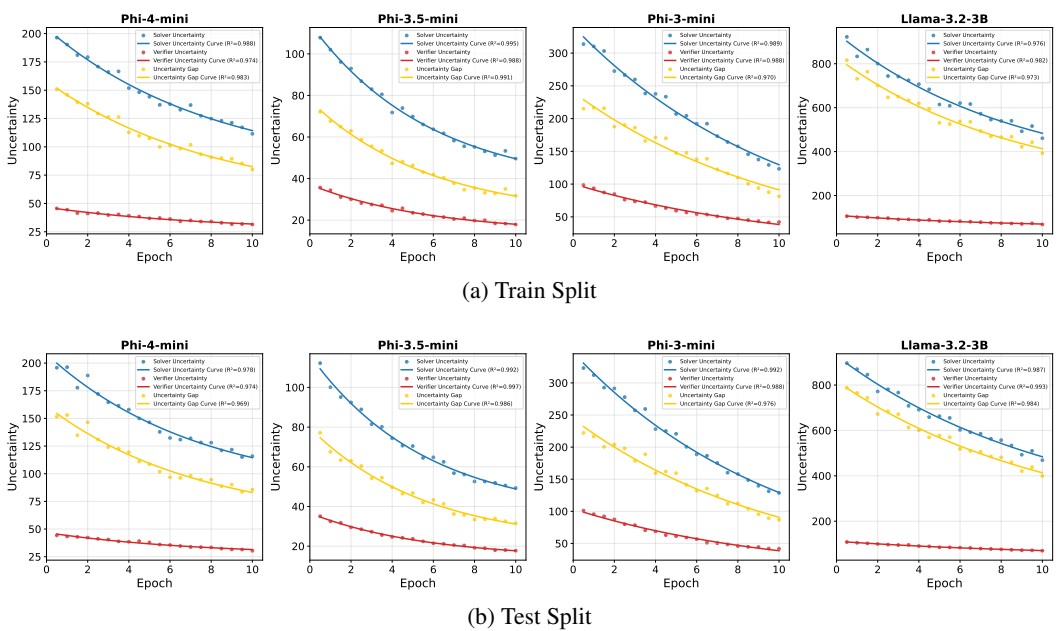

Figure 16: Uncertainty curve versus epochs of Phi-4-mini, Phi-3.5-mini, Phi-3-mini and Llama-3.2-3B models using TF method on ProntoQA datasets.

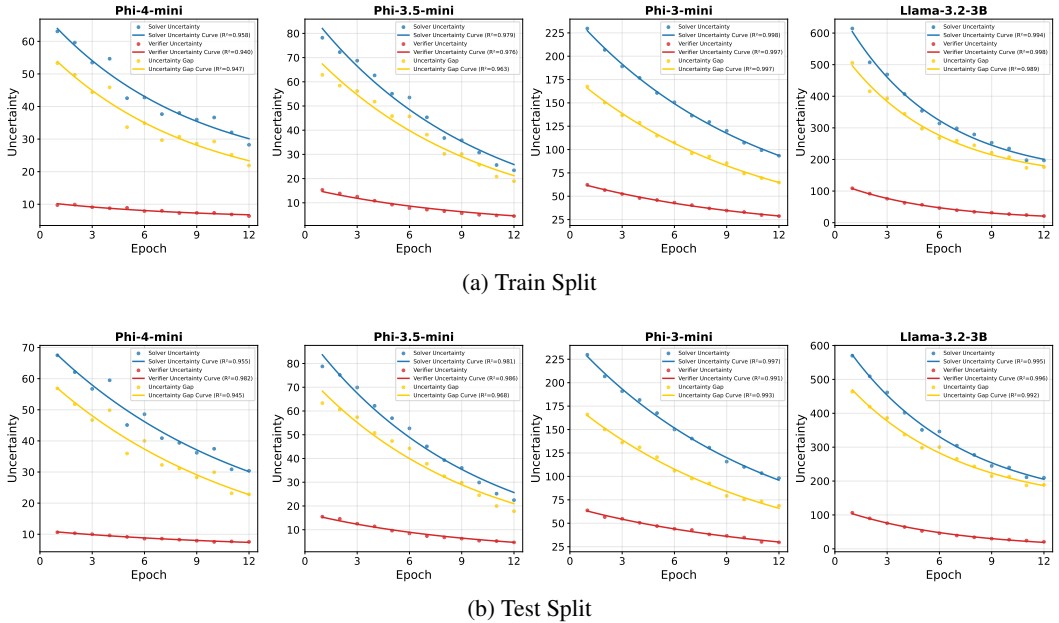

Figure 17: Uncertainty curve versus epochs of Phi-4-mini, Phi-3.5-mini, Phi-3-mini and Llama-3.2-3B models using TF method on MBPP datasets.

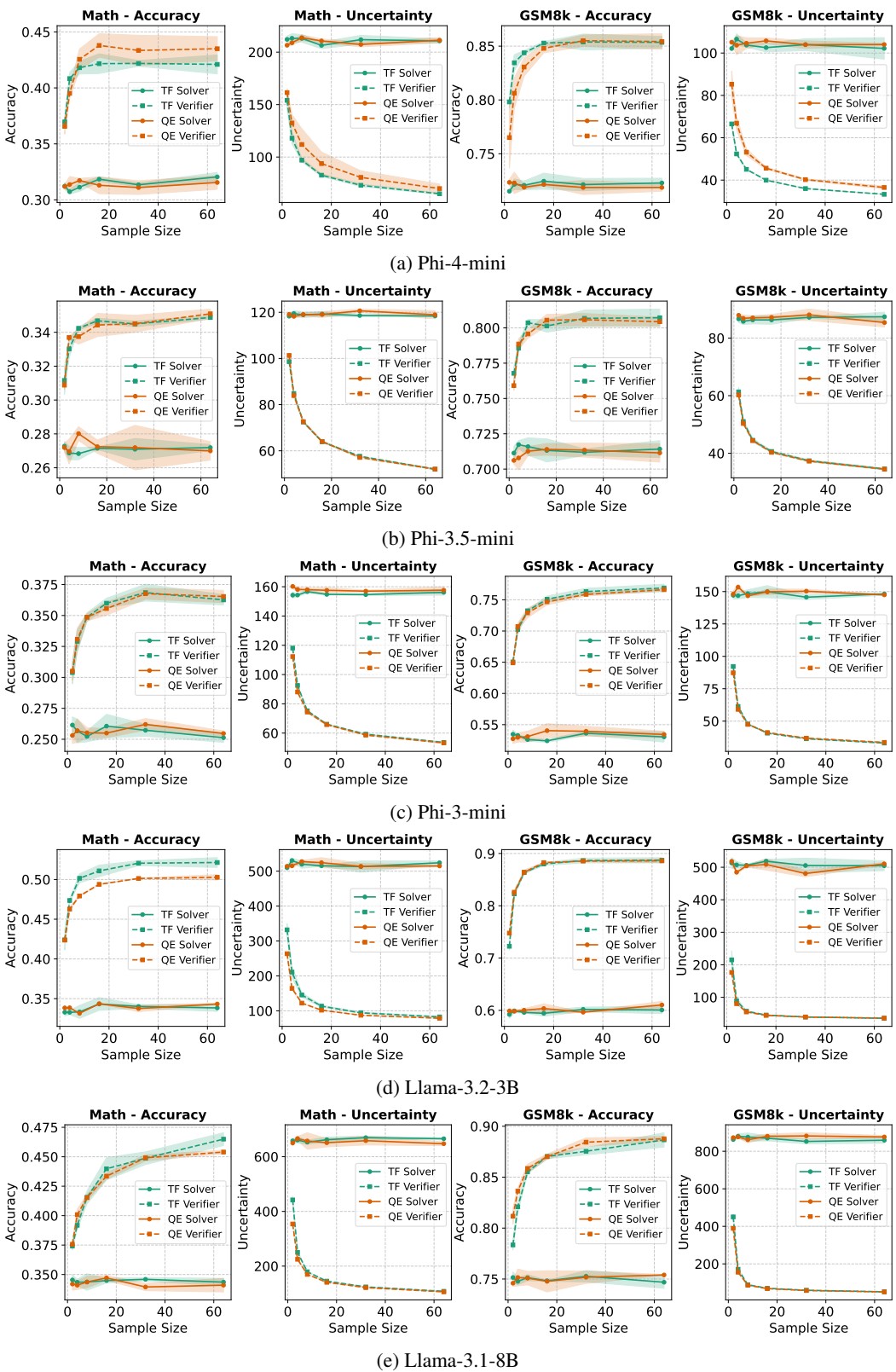

Figure 18: Accuracy and uncertainty of Phi-4-mini, Phi-3.5-mini, Phi-3-mini, Llama-3.2-3B and Llama-3.1-8B on Math and GSM8k across different sample size using TF and QE respectively.

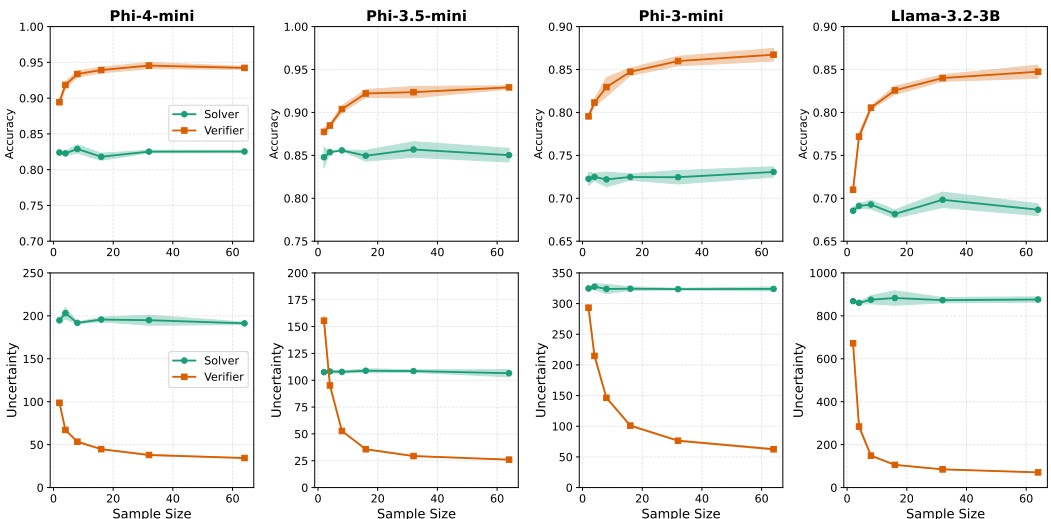

Figure 19: Accuracy and uncertainty of Phi-4-mini, Phi-3.5-mini, Phi-3-mini and Llama-3.2-3B on ProntoQA dataset across different sample size using TF method.

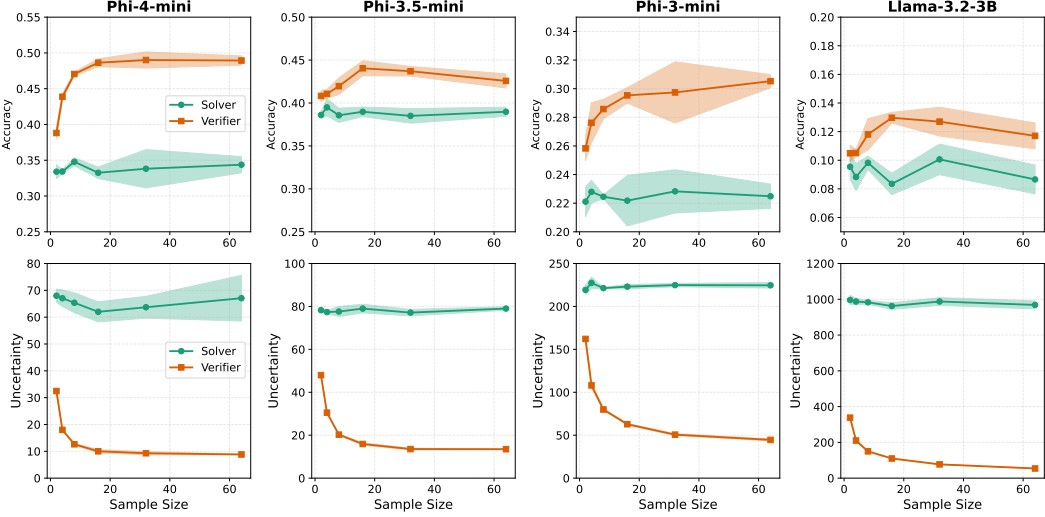

Figure 20: Accuracy and uncertainty of Phi-4-mini, Phi-3.5-mini, Phi-3-mini and Llama-3.2-3B on MBPP dataset across different sample size using TF method.

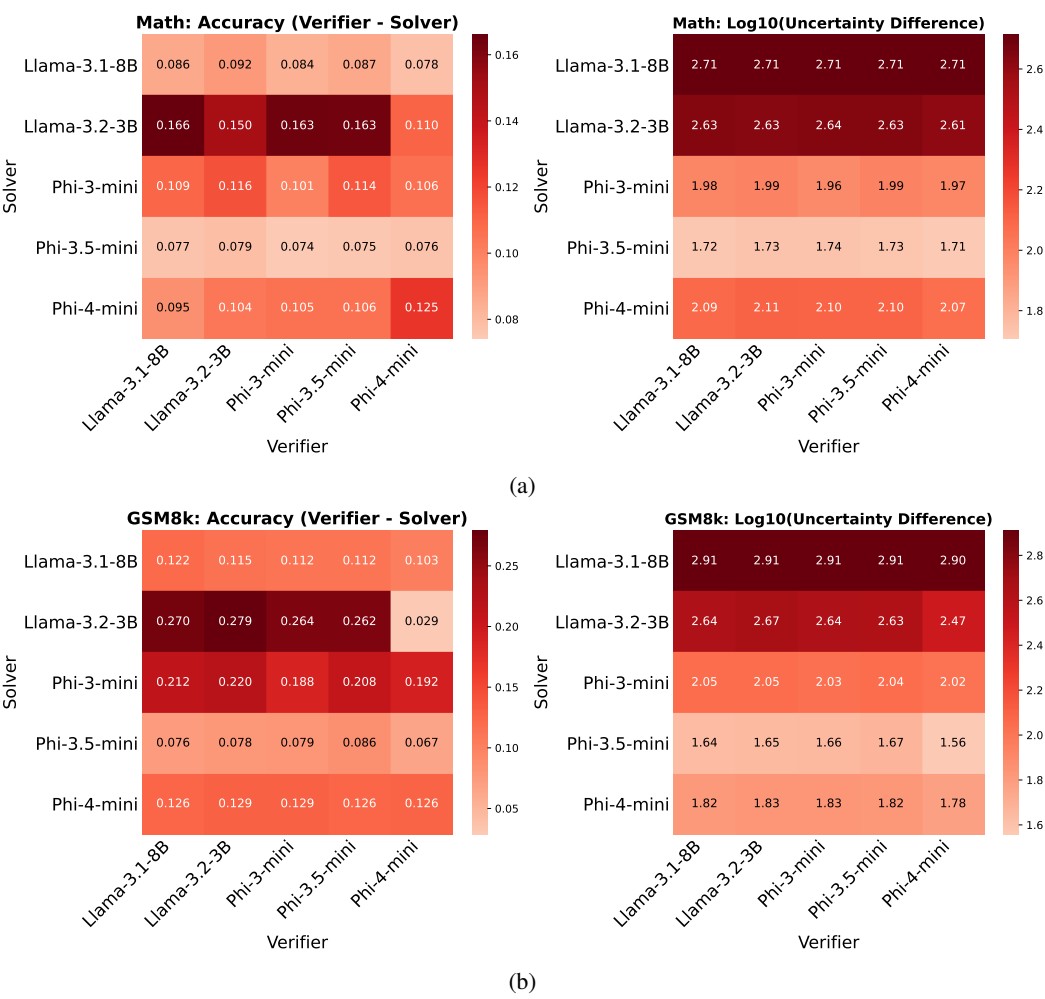

Figure 21: (a) Cross-evaluation using QE method with sample size N=16 on Math dataset. (b) Cross-evaluation using QE method with sample size N=16 on GSM8k dataset. For each solver model, sampled responses are verified by different models.

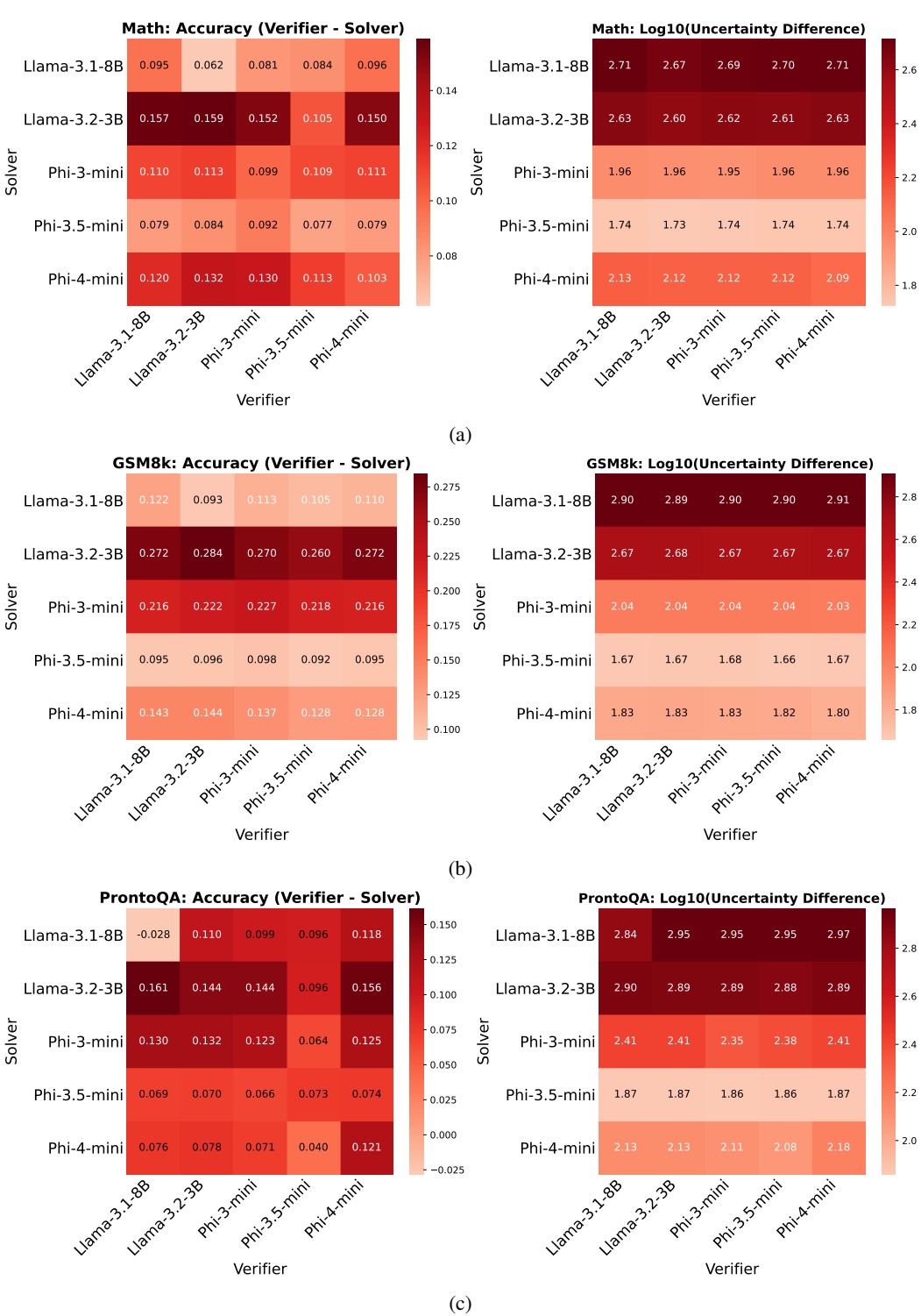

Figure 22: (a) Cross-evaluation using TF method with sample size N=16 on Math dataset. (b) Cross-evaluation using TF method with sample size N=16 on GSM8k dataset. (c) Cross-evaluation using TF method with sample size N=16 on ProntoQA dataset. For each solver model, sampled responses are verified by different models. Figures on the left illustrate the difference of accuracy while figures on the right show the 10 logarithms of the uncertainty difference.

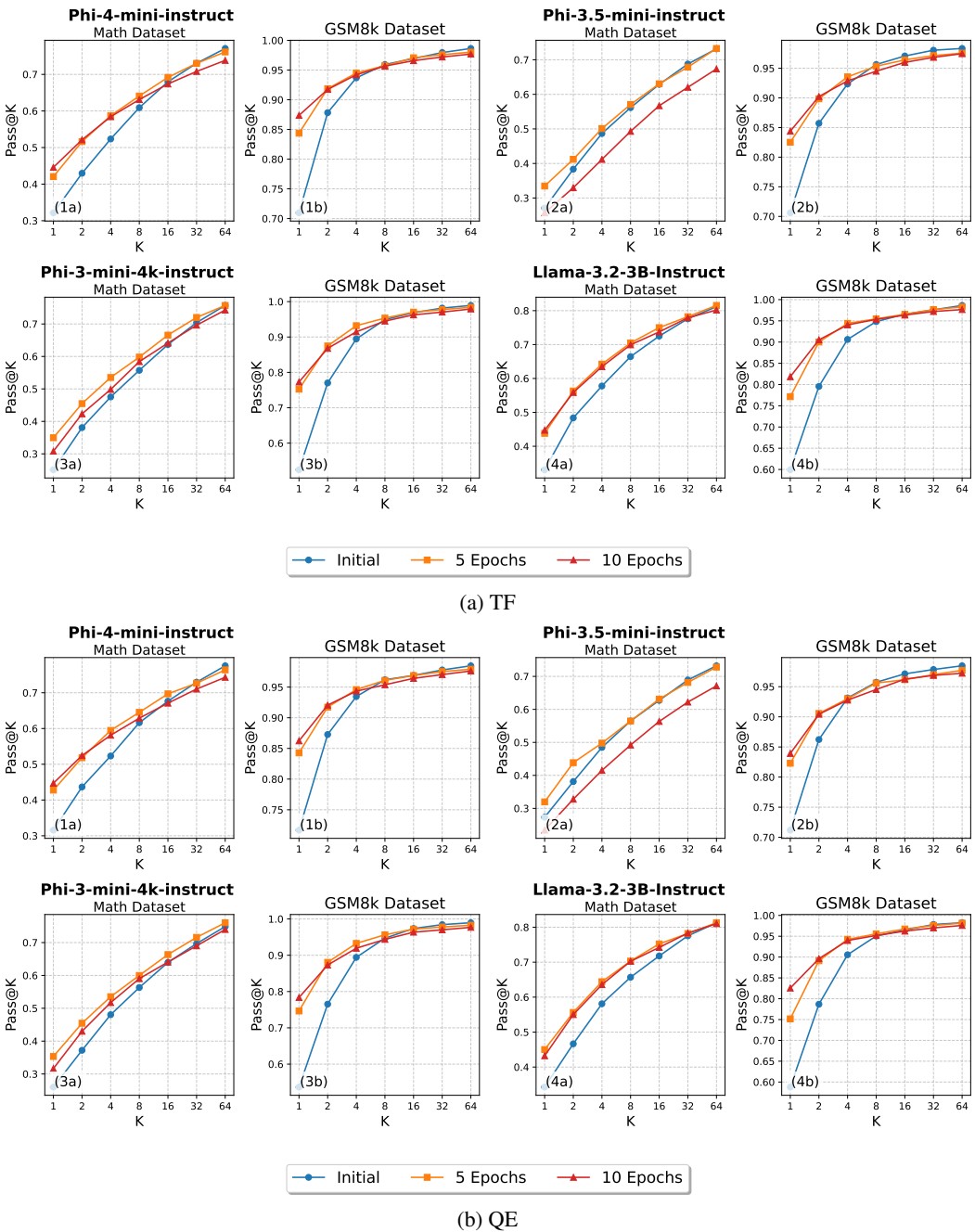

Figure 23: Pass@K with TF and QE method for different $K$ at $t = 0$, $t = 5$ and $t = 10$. Pass@K evaluates the diversity of model generation. The efficacy of the self-improvement process is demonstrated by an increase in the Pass@K metric for small values of $K$. Conversely, for large values of $K$, a decrease in Pass@K is observed. This phenomenon suggests that the diversity of generations decreases during the self-improvement process.

# E    ADDITIONAL DISCUSSION

**Phenomenological results.** We acknowledge that neural network optimization is complex. However, our approach is phenomenological, akin to how thermodynamics describes complex molecular systems using macroscopic variables. Our core assumption, the linear approximation of potential energy is empirically validated. As shown in Figure 2 and Figure 7, the relationship between the gap $G$ and its rate of change $dG/dt$ is strictly linear, with $R^2 > 0.9$. Based on linear assumption, exponential convergence law (Eq 10 & Eq 11) fits the training trajectories shown in Figure 4, with $R^2 > 0.9$. Although deriving this relationship from first principles remains an open challenge, our framework yields valuable insights into the self-improvement mechanism, it provides a precise mathematical characterization of the optimization curve and a rigorous basis for modeling cross-improvement strategies.

**Simplified assumptions (linear and time-invariant).** We clarify that linearity is inspired by empirical observations, not just a prior assumption, and the time-invariance is a local approximation. Figures 2 and 7 plot the $dG/dt$ against $G(t)$, revealing a linear relationship between the potential energy $E(t)$ and gap $G(t)$ across various models.

- **Model Complexity and Parameter Estimation.** Modeling the coefficients $(\alpha, \beta)$ as time-varying functions would significantly increase the complexity of the theoretical framework. Introducing time-varying parameters would result in complex modeling processes and make parameter estimation nearly impossible without introducing further arbitrary assumptions. Given that our constant-coefficient model already achieves a high degree of precision (with $R^2 > 0.9$ in fitting training dynamics), suggesting the time-invariant assumption is the most effective approach for capturing the macroscopic dynamics.

- **Time-Homogeneity of the State.** In our framework, the dynamics are driven by the state of the system (the Solver-Verifier Gap) rather than explicit time. This implies time-homogeneity: any time point $t$ in the training process can be regarded as a start point. The evolution of the system depends on the current capability gap $G(t)$ and the potential energy $E(t)$. This is empirically supported by Figure 2, which shows a consistent linear relationship between the gap and its rate of change throughout the training process, confirming that the parameters governing this relationship remain stable.

- **Applicability of Time-Varying Parameters.** We acknowledge that the time-invariant assumption is a local approximation valid for the standard self-improvement regimes studied in this paper. In specific complex scenarios, such as training across radically different data distributions or using highly dynamic learning rate schedules, the coefficients might drift. We have identified relaxing the time-invariant property as a specific direction (e.g., piece-wise constant) for future work to extend the framework's generality.

**Connection between theoretical variables and practical LLM learning signals.** The variables in our phenomenological model map directly to concrete, measurable quantities in the LLM training process:

- *The capability gap $G(t)$:* As defined in $Eq.(6)$, $G(t) = U_v(t) - U_s(t)$. This represents the "Quality Delta" between the model's current generation performance $U_s$, and the quality of the verification output $U_v$. In practice, this is measured by comparing the average NLL of the model's outputs against the average NLL of the Best-of-N responses selected for training.

- *The potential energy $E(t)$:* E(t) is a function of the gap, approximated linearly as $E(t) = kG(t) - b$. This represents the power which drives self-improvement. Equation 8 states $\frac{dU_s(t)}{dt} = -\alpha E(t)$. This means $E(t)$ maps directly to the speed of improvement of the model. In a practical training run, a high $E(t)$ indicates that the gap is large enough to provide strong gradients, leading to a rapid drop in loss. A low $E(t)$ indicates the model has exhausted the information in the current gap, leading to a training plateau.

Therefore, monitoring $E(t)$ and $G(t)$ might be equivalent to monitoring the efficiency of the data relative to the model's current state.

**Validations of cross-improvement (Equation 12).** We justify its validity from both theoretical and empirical perspectives:

- *Theoretical Rationality:* Equation 12 models the capability enhancement as a gain factor mechanism. The term $(1+\gamma\eta_t)^{-1}$ acts as a discount factor on the uncertainty. As the ratio of high-quality external data $\eta_t$ increases, the verifier's uncertainty $U_v$ decreases monotonically. This phenomenological formula captures the intuition that external data reduce the system's entropy, with $\gamma$ representing the efficiency of this external information. From a technical standpoint, we explicitly chose this functional form for its mathematical tractability. This specific inverse-linear form allows the coupled dynamics to be approximated by linear matrix operations. If we were to use more complex non-linear scaling laws, the system would become analytically intractable, preventing the derivation of the closed-form approximate solution in Proposition 5.1. This form strikes a necessary balance between capturing physical intuition and maintaining solvability.

- *Empirical Validation:* Crucially, the validity of this modeling assumption is supported by our experimental results. Based on Eq.12, we derived Proposition 5.1, which predicts that the final solver capability depends primarily on the total amount of external data, rather than its specific allocation timing. Our experiments in Section 5.2 (Table 1) confirm this conclusion. We observed that different allocation strategies (Early, Uniform, Late) yielded negligible differences in final accuracy. This alignment between the theoretical prediction derived from Eq.12 and the empirical outcome strongly validates the reasonableness of the model.

**Results and difference of model families.** Different model families exhibit distinct exponential trajectories because their inherent properties (such as pre-training data quality and architectural maturity) determine the specific values of the decay rate coefficient and the boundary conditions in our derived analytical solution. As discussed in Q1, Llama families exhibit a larger solver learning rate $\alpha$. Mathematically, a larger $\alpha$ maximizes the exponent magnitude $|-k(\alpha-\beta)|$, resulting in a steeper decay curve (rapid convergence), as observed in our Llama experiments. Phi families are pre-trained on high-quality synthetic data. This results in a smaller gap and decay constant, leading to a flatter decay function. Crucially, the decay function is mathematically coupled to the limit. The theoretical asymptote is defined as $U_{s,\infty} = U_{s,0} - \frac{\alpha}{\alpha-\beta}(G_0 - \frac{b}{k})$.

- *Phi Family:* Starts with a significantly smaller Initial Gap ($G_0$) and lower initial uncertainty ($U_{s,0}$) due to high-quality upstream synthesis. Plugging these smaller values into the equation mathematically dictates a better final capability.

- *Llama Family:* Starts with a larger $G_0$ and $U_{s,0}$. Despite the rapid initial descent, the higher starting baseline mathematically leads to convergence at a relatively higher asymptote.

**Potential energy as driven force.** We justify modeling the training dynamics as Equation (9):

- *Physical Analogy: Potential Difference as the Driving Force.* In this paper, we model the training dynamics as a form of $\frac{dU_s}{dt} = -\alpha E(t), \frac{dU_v}{dt} = -\beta E(t)$, which can be analogous to Newton's Second Law. In Newton's Second Law ($F = ma$), force is the cause that generates acceleration. Without force, there is no change in the state of motion. Similarly, in our framework, the Potential Energy $E(t)$ acts as the force which is the cause that drives the evolution of the model's capabilities ($U_s, U_v$)(inspired by Song et al. (2025)). Just as a physical force dictates how an object moves, the magnitude of this potential energy dictates the intensity of the self-improvement process. This modeling captures the core intuition: the existence of a gap creates a potential difference that physically necessitates and drives the optimization process.

- *Empirical Validation.* Despite the simplicity of our model, our results demonstrate that this simple macroscopic model captures the training dynamics with remarkable precision. We fit the theoretical exponential curves derived from our conservative field model to the actual training trajectories. As shown in Figure 4, despite the model's simplicity, it fits the empirical training data (both Math and GSM8k datasets) exceptionally well, with coefficients of determination ($R^2$) consistently exceeding 0.9.

**Conservative vector field.** To statistically verify whether historical information ($G(t-1)$) provides any necessary explanatory power for the dynamics, we performed a Partial F-test to compare two nested regression models:

- Model 1 (Simple Model): $\frac{dG}{dt} = \beta_1 G(t) + \epsilon$
- Model 2 (Full Model): $\frac{dG}{dt} = \beta_1 G(t) + \beta_2 G(t-1) + \epsilon$

We evaluated whether adding the historical term $G(t-1)$ significantly reduces the Residual Sum of Squares (RSS). As shown in Table 4, the insignificant F-test result leads us to fail to reject the null hypothesis ($\beta_2 = 0$). This statistically proves that adding historical information does not significantly improve the model's fit. The dynamics are adequately described by the current state alone, providing robust evidence for the memoryless nature of the potential energy.

Table 4: Model Selection via Partial F-test.

| Model | Dataset | Split | F-statistic | $P$-value (F) |
|---|---|---|---|---|
| **Phi-4-mini** | Math | Train | 1.295 | 0.2687 |
| | Math | Test | 1.203 | 0.2911 |
| | GSM8k | Train | 0.024 | 0.8771 |
| | GSM8k | Test | 9.277 | 0.0122 |
| **Llama-3.2-3B** | Math | Train | 0.479 | 0.5114 |
| | Math | Test | 0.833 | 0.3917 |
| | GSM8k | Train | 0.875 | 0.3808 |
| | GSM8k | Test | 0.808 | 0.3985 |
| **Phi-3.5-mini** | Math | Train | 1.807 | 0.2209 |
| | Math | Test | 3.218 | 0.1159 |
| | GSM8k | Train | 1.945 | 0.2058 |
| | GSM8k | Test | 3.140 | 0.1197 |
| **Phi-3-mini** | Math | Train | 0.000 | 0.9845 |
| | Math | Test | 5.300 | 0.0548 |
| | GSM8k | Train | 0.528 | 0.4911 |
| | GSM8k | Test | 2.163 | 0.1848 |

