# OpenReview forum: "Theoretical Modeling of Large Language Model Self-Improvement Training Dynamics Through Solver-Verifier Gap"
_ICLR.cc/2026/Conference — ICLR 2026 Poster_

### Official Review · Reviewer_V9jn · 2025-10-19

**Soundness:** 3
**Presentation:** 3
**Contribution:** 3
**Rating:** 8
**Confidence:** 2

**Summary:**

The paper develops a physics-inspired theoretical framework to explain how large language models (LLMs) improve themselves without external data. It introduces the concept of a solver-verifier gap—the difference between an LLM’s ability to generate solutions (solver) and its ability to evaluate them (verifier)—as the main driver of self-improvement. The authors model training dynamics using coupled differential equations, showing that solver and verifier capabilities evolve exponentially toward limits determined by their initial gap. Experiments on Phi and Llama models across multiple datasets validate these exponential laws and confirm that verifiers consistently outperform solvers. The paper further extends the framework to cross-improvement, where limited external data enhance verifier capability; theoretical and empirical results show that the timing of using such data has minimal impact on final performance. Overall, the study provides a unified quantitative model linking LLM self-improvement behavior, convergence limits, and external data effects

**Strengths:**

This paper is highly original in formalizing LLM self-improvement through a physics-inspired solver-verifier gap framework, transforming an empirical phenomenon into a quantitative model. Its quality is strong, with clear theoretical derivations, well-controlled experiments, and convincing exponential fits that validate the framework across multiple models and datasets. The writing is clear and logically structured, effectively integrating intuition, math, and visualization. In significance, the work provides a foundational step toward understanding autonomous model improvement and offers practical insights for optimizing self- and cross-improvement strategies, making it both conceptually innovative and broadly impactful for future LLM research.

**Weaknesses:**

Some claims appear to be a bit strong - "Self-improvement is among the most prominent techniques within the realm of ...".

I also advocate for using the term "generation-verification" gap to respect the originality.

The time-invariant assumption sounds a bit strong: this assumes that the key parameters governing training dynamics — such as the cross-improvement effect γ, and the solver/verifier rate coefficients α and β remain constant across all epochs. Some more ablations would be helpful to understand their effects.

Some limiting behaviors are worthy of examination - such as how γ α and β would affect the convergence.

In EvoLM, model developers have found that base models that have undergone mid-training can be more easily adaptable using RL. The work should take into account such factors and see how this affects the self-improvement behaviors. Ideally, a controlled comparison using base, mid, post-trained checkpoints would be an insightful add. Because the field of post-training can heavily depend on upstream training, I am leaning towards advocating for strong accept if this confounder is taken into account.

**Questions:**

How does the base model FLOPs (and families) affect the analysis?

how do models from different families have different exponential decay functions, which would lead to the modeling of solver and verifier uncertainties (or their gaps) toward asymptotic limits?

---

> ### Author Response · Authors · 2025-11-23
> **Clarifications and additional experiments**
>
> We express our sincere gratitude to you for the positive assessment and for recognizing our work's originality and foundational significance. We are particularly grateful for your insightful suggestion regarding the confounder of upstream training stages. This specific guidance motivated us to conduct a new set of controlled experiments, which helps deepens our understanding of model dynamics. We have carefully addressed this key point along with your suggestions on terminology and assumptions, and we hope these revisions merit your strong advocacy for acceptance.
>
> >Some claims appear to be a bit strong - "Self-improvement is among the most prominent techniques within the realm of ...".
>
> Thanks for pointing it out! We have revised the manuscript to state: "Self-improvement is a significant technique..." instead of "most prominent."
>
> > I also advocate for using the term "generation-verification" gap to respect the originality.
>
>  We apologize for any confusion that might cause. We acknowledge the priority of Song et al. (2025) in conceptualizing the gap. We have added a footnote explicitly linking our "solver-verifier" gap to the "generation-verification" gap to respect their originality, while clarifying that we use "solver-verifier" to align with our specific Best-of-N implementation definitions.
> % We use this term at the time being due to the definition of the solver and verifier in the main text.
>
> >The time-invariant assumption for $\alpha,\beta,\gamma$ sounds strong.
>
> We clarify that time-invariance is a local approximation, similar to assuming constant acceleration in short-term mechanics. While training dynamics are complex, the high empirical fit shown in Figure 4 and Figure 11 confirms that this approximation effectively captures the macroscopic trajectory during the SFT.
> We leave the time-variant version (with potentially different constraints) for future work.
>
> > Limiting behaviors are worthy of examination - such as how $\alpha,\beta,\gamma$ affect convergence.
>
>  Thank you for pointing it out! Eq.10 shows the convergence speed is determined by the difference between solver and verifier learning rates. A larger $\alpha$ accelerates convergence. Corollary 3.1 and Eq.11 show that the final capability limit is determined by the initial gap $G_0$ and the ratio $\beta/(\alpha-\beta)$. Proposition 5.1 demonstrates that $\gamma$ effectively scales the verifier potential, raising the ceiling for the solver's ultimate performance.
>
> >Ideally, a controlled comparison using base, mid, post-trained checkpoints would be an insightful add.
>
>  We fully agree that disentangling the impact of upstream training is critical. To address this confounder, we have launched a controlled experiment using the EvoLM 1B model suite [Qi et al., 2025]. We compare the self-improvement dynamics of the following three checkpoints on the GSM8k datasets: SFT Baseline (1B-160BT-100Kep1), Mid-train model (1B-160BT-8+42BT-100Kep1) and Post-trained model (1B-160BT-8+42BT-100Kep1-100Kep16). We present the results in Figure 22 and Figure 23. By fitting our differential equations to the training trajectories, we identified three distinct dynamical regimes governed by model training stage. The Standard SFT Baseline exhibits the highest solver learning rate, which represents high-plasticity. The Mid-train Enhanced model improves the initial capability. The Post-trained model exhibits the lowest learning rate and a flattened trajectory. This confirm a fundamental law of self-improvement dynamics: as model maturity increases (from Base to Post-trained), the potential for further self-improvement diminishes.

---

> > ### Author Response · Authors · 2025-11-23
> > **Clarifications and additional experiments (2)**
> >
> > > How does the base model FLOPs (and families) affect the analysis?
> >
> >  Base model characteristics (FLOPs and families) do not change the form of the governing differential equations (Eq.1), but they significantly shift the coefficients ($\alpha,\beta$) and initial gap ($G_0$).
> > 1. **Impact of FLOPs:** To address how FLOPs affect the analysis, we conducted a comparative experiment between EvoLM-1B and EvoLM-4B. The results show that larger models exhibit significantly lower initial and asymptotic uncertainty. Meanwhile, larger models demonstrate a higher exponential decay rate. Scaling up the model size creates more optimization space, which allows the solver to convert the gap into capability gains more efficiently.
> > 2. **Impact of Model Families:** Phi families are trained on high-quality synthetic data. This process acts as a form of upstream distillation, where the model internalizes the reasoning patterns of a stronger teacher model during pre-training. As a result, Phi models have a lower initial gap $G_0$ and $\alpha$. Llama models are typically trained on massive amounts of raw web tokens. Consequently, the Llama model starts with lower initial performance and a larger Initial Gap ($G_0$) and $\alpha$.
> >
> > > How do models from different families have different exponential decay functions, which would lead to the modeling of solver and verifier uncertainties (or their gaps) toward asymptotic limits?
> >
> >  Different model families exhibit distinct exponential trajectories because their inherent properties (such as pre-training data quality and architectural maturity) determine the specific values of the decay rate coefficient and the boundary conditions in our derived analytical solution. As discussed in Q1, Llama families exhibit a larger solver learning rate $\alpha$. Mathematically, a larger $\alpha$ maximizes the exponent magnitude $|-k(\alpha-\beta)|$, resulting in a steeper decay curve (rapid convergence), as observed in our Llama experiments. Phi families are pre-trained on high-quality synthetic data. This results in a smaller gap and decay constant, leading to a flatter decay function. Crucially, the decay function is mathematically coupled to the limit. The theoretical asymptote is defined as $U_{s,\infty}=U_{s,0}-\frac{\alpha}{\alpha-\beta}(G_0-\frac{b}{k})$.
> > 1. **Phi Family:** Starts with a significantly smaller Initial Gap ($G_0$) and lower initial uncertainty ($U_{s,0}$) due to high-quality upstream synthesis. Plugging these smaller values into the equation mathematically dictates a better final capability.
> > 2. **Llama Family:** Starts with a larger $G_0$ and $U_{s,0}$. Despite the rapid initial descent, the higher starting baseline mathematically leads to convergence at a relatively higher asymptote.
> >
> > We sincerely thank you again for your constructive suggestions, which have been instrumental in improving the quality and clarity of our work. We have carefully addressed your questions in the responses above and incorporated the corresponding revisions into the updated manuscript (highlighted in blue). We are eager to provide any further clarifications to help your evaluation!

---

### Official Review · Reviewer_Sog4 · 2025-10-30

**Soundness:** 2
**Presentation:** 2
**Contribution:** 2
**Rating:** 4
**Confidence:** 3

**Summary:**

The paper investigates the training dynamics underlying the self-improvement process of large language models, aiming to uncover the mechanisms driving performance evolution without relying on external data. The authors propose a theoretical framework based on the solver–verifier gap, which explains that the essential source of self-improvement lies in the disparity between a model’s generation (solver) and self-evaluation (verifier) capabilities. Inspired by the potential energy framework in physics, the authors formulate a set of coupled differential equations to characterize how the solver and verifier capabilities evolve over time (i.e., training iterations), and derive that their capability growth follows an exponential convergence pattern.

**Strengths:**

1. The paper provides an interesting and effective theoretical exposition of LLM self-improvement, accompanied by a relatively rigorous theoretical proof.

2. It conducts diverse empirical studies on mathematical tasks across multiple models, which enhances the credibility of the results.

**Weaknesses:**

1. The relationship between model capability and output uncertainty is not clearly articulated. The connection between a model’s accuracy on different tasks, its output uncertainty, and its underlying “capability” remains ambiguous. Why can output uncertainty serve as a valid indicator of model capability? How is this metric related to output accuracy? Although the authors provide some explanation in the appendix, the paper lacks further theoretical and empirical justification for the appropriateness of this metric.

2. The theoretical analysis relies on numerous and rather simplified assumptions. In particular, the energy-based assumptions regarding the training process may fail to capture the complex dynamics involved in model optimisation.

3. The experiments are limited to mathematical reasoning tasks (GSM8K, Math) and lack broader validation on code generation or other reasoning benchmarks.

**Questions:**

Please refer to the relevant points in the Weaknesses section. If the authors can provide clarification and improvements, I would be very happy to raise my score.

---

> ### Author Response · Authors · 2025-11-23
> **Clarifications and additional experiments**
>
> Thank you for your thoughtful and constructive feedback. We are encouraged that you recognize our theoretical framework as "interesting and effective" and our proofs as "relatively rigorous."
>
> We have carefully engaged with your concerns regarding the justification of the uncertainty metric, the modeling assumptions, and the experimental scope. Below, we provide our detailed answers for your concerns.
>
> >Why can output uncertainty serve as a valid indicator of model capability?
>
> In this paper, we use uncertainty instead of accuracy as a proxy metric for model capability. The justifications are detailed as follows:
> 1. **Sharpening Mechanism:** Following Huang et al. (2025), minimizing the loss ($U_v$) forces the model to sharpen its intrinsic probability distribution onto high-quality responses. A decrease in $U_s$ signifies that this distributional shift has occurred and that the model has successfully learned from the verifier's output. Thus, $U_s$ is the direct measure of the model's capability.
> 2. **Empirically correlation between accuracy and uncertainty:** To validate that $U_s$ indicates the model's capability, we calculate the Pearson correlation coefficient (r) between $U_s$ and solver accuracy for different models and datasets. Accuracy and Uncertainty for most model-dataset pairs are significantly negatively correlated ($|r|>0.8$), which proves that reducing solver uncertainty ($U_s$) linearly translates to gains in accuracy.
> 3. **Absence of Ground Truth:** In many real-world scenarios, particularly open-ended generation tasks, ground-truth labels are often unavailable or difficult to define. Unlike accuracy, which strictly depends on external gold references, uncertainty is computed from the model's output distribution. This makes uncertainty a necessary and universal metric for monitoring capability evolution in label-free regimes where accuracy calculation is infeasible.
> 4. **The Optimization Objective:** In standard SFT, the objective of training is to maximize the likelihood of correct responses, which is mathematically equivalent to minimizing their uncertainty (Negative Log-Likelihood). In our framework, we regard the output of the verifier as the ground truth for SFT. The loss function at step $t$ is defined as $L_t(f) \triangleq U_v(t) =-\frac{1}{n}\sum_{i=1}^{n}\log\ \pi_f(\hat{y}_{i}^{\text{BoN}}(t)|x_i)$, which is identical to our definition of verifier uncertainty $U_v(t)$.
>
> We have added this section to the Appendix A.3, and we appreciate your suggestion which has improved our paper.
>
> >The theoretical analysis relies on simplified assumptions (energy-based) which may fail to capture complex dynamics.
>
> We acknowledge that neural network optimization is complex. However, our approach is phenomenological, akin to how thermodynamics describes complex molecular systems using macroscopic variables. Our core assumption, the linear approximation of potential energy is empirically validated. As shown in Figure $2$ and Figure $7$, the relationship between the gap $G$ and its rate of change $dG/dt$ is strictly linear, with $R^2>0.9$. Based on linear assumption, exponential convergence law (Eq 10 \& Eq 11) fits the training trajectories shown in Figure 4, with $R^2>0.9$. Although deriving this relationship from first principles remains an open challenge, our framework yields valuable insights into the self-improvement mechanism, it provides a precise mathematical characterization of the optimization curve and a rigorous basis for modeling cross-improvement strategies.
>
> >The experiments are limited to mathematical reasoning tasks and lack broader validation on code generation or other reasoning benchmarks.
>
> We fully agree that validating the framework on diverse domains is essential. To address this, we conducted additional experiments on two distinct benchmarks:
> 1. **MBPP:** A benchmark representing code generation capabilities.
> 2. **ProntoQA:** A dataset focused on deductive logical reasoning.
>
> Detailed results are presented in Appendix D. Consistent with our findings on mathematical tasks, Figure 10 and Figure 11 demonstrate that both solver and verifier capabilities consistently improve during the self-improvement process on these new domains. Crucially, the validation in Figure 14 and Figure 15 confirms that the training dynamics for both Code and Logic tasks strictly align with the exponential convergence law proposed in our theoretical framework ($R^2 > 0.9$). These results strongly support that our findings are not limited to mathematics but generalize effectively to broader reasoning tasks.
>
> We sincerely thank you again for your constructive suggestions, which have been instrumental in improving the quality and clarity of our work. We have carefully addressed your questions in the responses above and incorporated the corresponding revisions into the updated manuscript (highlighted in blue).
> We are eager to provide any further clarifications to help your evaluation!

---

> > ### Comment · Reviewer_Sog4 · 2025-11-28
> >
> > Thank you very much for the author’s detailed reply. I no longer have concerns regarding output uncertainty as a valid measure of model capability, nor about the generality of the framework, and I will raise my rating of the paper accordingly — although it is rather odd that the current rating cannot be edited.
> >
> > At the same time, I still have questions regarding the assumptions about the system’s potential energy in the paper. Why can the training dynamics of a language model be modeled as a conservative vector field, and how can we ensure that the energy differences between different capabilities remain constant during training (i.e., that the energy of a state depends only on the state itself)? This perhaps requires a clearer explanation, even though the assumption of potential energy in the original text does not compromise the rigor of the subsequent derivations.
> >
> > If the author could help clarify these points, I would be very willing to further increase my rating.

---

> > > ### Author Response · Authors · 2025-11-28
> > > **Further clarifications on  the assumptions about the potential energy**
> > >
> > > We sincerely appreciate your constructive and professional suggestions. We are happy that our clarifications have successfully resolved the concerns regarding output uncertainty as a capability and the generality of the framework. We value this opportunity to further elaborate on the assumptions underlying the potential energy model. We address this concern point-by-point below:
> > >
> > > > Why can the training dynamics of a language model be modeled as a conservative vector field?
> > >
> > > We justify modeling the training dynamics as a conservative vector field from two perspectives:
> > > 1. **Physical Analogy: Potential Difference as the Driving Force.** In this paper, we model the training dynamics as a form of $\frac{dU_s}{dt}=-\alpha E(t),\frac{dU_v}{dt}=-\beta E(t)$, which can be analogous to Newton's Second Law.  In Newton's Second Law ($F=ma$), force is the cause that generates acceleration. Without force, there is no change in the state of motion. Similarly, in our framework, the Potential Energy $E(t)$ acts as the force which is the cause that drives the evolution of the model's capabilities ($U_s,U_v$) (inspired by Song et al. (2025)). Just as a physical force dictates how an object moves, the magnitude of this potential energy dictates the intensity of the self-improvement process. This modeling captures the core intuition: the existence of a gap creates a potential difference that physically necessitates and drives the optimization process.
> > > 2. **Empirical Validation.** Despite the simplicity of our model, our results demonstrate that this simple macroscopic model captures the training dynamics with remarkable precision. We fit the theoretical exponential curves derived from our conservative field model to the actual training trajectories. As shown in Figure 4, despite the model's simplicity, it fits the empirical training data (both Math and GSM8k datasets) exceptionally well, with coefficients of determination ($R^2$) consistently exceeding $0.9$.
> > >
> > > >how can we ensure that the energy differences between different capabilities remain constant during training (i.e., that the energy of a state depends only on the state itself)?
> > >
> > > This concern is around why the coefficients $\alpha$ and $\beta$ in our framework are modeled as time-invariant constants (please point it out if we misunderstand this question, thanks!). We justify this modeling choice from three perspectives:
> > >
> > > 1. **Model Complexity and Parameter Estimation.** Modeling the coefficients ($\alpha,\beta$) as time-varying functions would significantly increase the complexity of the theoretical framework. Introducing time-varying parameters would result in complex modeling processes and make parameter estimation nearly impossible without introducing further arbitrary assumptions. Given that our constant-coefficient model already achieves a high degree of precision (with $R^2>0.9$ in fitting training dynamics), suggesting the time-invariant assumption is the most effective approach for capturing the macroscopic dynamics.
> > > 2. **Time-Homogeneity of the State.** In our framework, the dynamics are driven by the state of the system (the Solver-Verifier Gap) rather than explicit time. This implies time-homogeneity: any time point $t$ in the training process can be regarded as a start point. The evolution of the system depends on the current capability gap $G(t)$ and the potential energy $E(t)$. This is empirically supported by Figure 2, which shows a consistent linear relationship between the gap and its rate of change throughout the training process, confirming that the parameters governing this relationship remain stable.
> > > 3. **Applicability of Time-Varying Parameters.** We acknowledge that the time-invariant assumption is a local approximation valid for the standard self-improvement regimes studied in this paper. In specific complex scenarios, such as training across radically different data distributions or using highly dynamic learning rate schedules, the coefficients might drift. We have identified relaxing the time-invariant property as a specific direction (e.g., piece-wise constant) for future work to extend the framework's generality.
> > >
> > > We once again appreciate your helpful and insightful suggestions. We remain active to provide any further clarifications to assist with your evaluation.

---

> ### Comment · Reviewer_Sog4 · 2025-11-28
>
> Thank you very much for your continued responses! The author’s explanation regarding the conservative vector field assumption indeed demonstrates thoughtful consideration, yet please allow me to maintain some differing opinions (which may stem from my initial question not being specific enough). From my perspective, the notion of potential energy differences serving as the driving force is itself derived from the conservative vector field assumption, and the subsequent theoretical developments are also based on this assumption. Therefore, in principle, the later derivations cannot be used as evidence supporting the validity of the conservative vector field assumption — although I fully acknowledge the impressive accuracy in capturing training dynamics in the later empirical validations.
>
> Secondly, I sincerely appreciate the author’s further clarification regarding time-invariant constants, though this was not the original intent of my question. However, these clarifications are indeed insightful. I hope the author can provide more explicit theoretical grounding for the conservative vector field assumption, or alternatively, a more localized experimental justification, so as to enhance the verifiability of this assumption. I believe such clarification will benefit the entire community, and I look forward to continuing this constructive discussion.

---

> > ### Author Response · Authors · 2025-11-29
> > **Further clarification on conservative vector field assumption**
> >
> > We deeply appreciate the rigorous scrutiny regarding the Conservative Vector Field assumption. We now understand that verifying this assumption requires proving that the improvement dynamics are driven by the current state (Gap) rather than historical trajectories. To statistically verify whether historical information ($G(t-1)$) provides any necessary explanatory power for the dynamics, we performed a Partial F-test to compare two nested regression models:
> > 1. **Model 1 (Simple Model):** $\frac{dG}{dt}=\beta_1 G(t)+\epsilon$
> > 2. **Model 2 (Full Model):** $\frac{dG}{dt}=\beta_1 G(t)+\beta_2 G(t-1) +\epsilon$
> >
> > We evaluated whether adding the historical term $G(t-1)$ significantly reduces the Residual Sum of Squares (RSS). As shown in Table 4, the insignificant F-test result leads us to fail to reject the null hypothesis ($\beta_2=0$). This statistically proves that adding historical information does not significantly improve the model's fit. The dynamics are adequately described by the current state alone, providing robust evidence for the memoryless nature of the potential energy.
> > We appreciate the reviewer's helpful and insightful suggestions again, which has greatly enhanced our manuscript. We have revised our manuscript accordingly (page 37).

---

### Official Review · Reviewer_ZsPe · 2025-11-01

**Soundness:** 2
**Presentation:** 2
**Contribution:** 3
**Rating:** 4
**Confidence:** 3

**Summary:**

The paper studies the training dynamics during LLM self-improvement training. The paper uses the solver-verifier gap as the key concept for measuring the self-improvement dynamics.  The solver-verifier gap is defined by the log ratio of policy's probability over the best of N generation and the individual generations. The paper proposes to model the generator's capability change and solver's capability change as a differential equation, and thus the change of gap can be measured as the energy function. Experiment results are provided to support the theoretical models. The paper further studies the dynamics of cross-improvement.

**Strengths:**

1. The paper studies the important problem of modeling the LLM self-improvement process, which models a very complex system with simple differential equations.

2. The paper provides experiments to verify the validity of the proposed models.

**Weaknesses:**

1. I had a difficult time even trying to understand the basic definitons such as solver uncertainty and verifier uncertainty. Are terms like $U_s(t)$ and $U_v(t)$ random variables, since $y_i$ are random variables? If so, how do we even understand these terms and so the gap term? How are they measured in practice?

2. I don't see the necessity of formulating everything as differential equations as the LLM self-improvement update is discrete.

3. The exeperiment setup seems unclear. For example, it is unclear to me what TF and QE methods are.

4. The result of the gap narrows during self-improvement does not seem new to me; it appeared in many previous works already.

5. The paper needs to highlight what are the significance of the current results.

**Questions:**

1. Although some experiments seem to suggest that $E(t)$ can be linear wrt $G(t)$, can the authors show if this is even possible in any toy settings, where we can even have the policy follows some bernoulli distribution parametrized by $p$, and we have some updating dynamics $d p(t)/dt$, can we say something that $E(t)$ is linear wrt $G(t)$?

2. Can the authors explain why does eq 12. a reasoning model?

3. In the final sentence of section 5.2, how did we reach the conclusion that the small difference in accuracy validates prop. 5.1?

---

> ### Author Response · Authors · 2025-11-23
> **Clarifications (1)**
>
> We sincerely thank you for the detailed feedback. We appreciate your recognition of our work's objective to model complex self-improvement dynamics using simple differential equations. We also deeply value your critical questions regarding the fundamental definitions and the necessity of the continuous framework. These comments have motivated us to significantly improve the clarity of our formalization and explicitly articulate the unique predictive significance of our quantitative results compared to prior qualitative observations. We address your concerns point-by-point below.
>
> >Are terms like $U_s$ and $U_v$ random variables, since $y_i$ are random variables?
>
> We apologize for the confusion. While a single generated response $y_i$ is a random variable, our metrics $U_s$ and $U_v$ are defined as the expectations (empirical averages) over the entire dataset.
> As formally defined in Equation 5, $U_s(t) \triangleq -\frac{1}{n}\sum_{i=1}^{n}\log\ \pi_f(\hat{y}_{i}(t)|x_i).$ By the Law of Large Numbers, calculating the average over a large number of prompts (n) stabilizes these values into deterministic scalars. This allows us to model their evolution using deterministic differential equations. In practice, we measure them by logging the average Negative Log-Likelihood (NLL) of the Solver's responses ($U_s$) and the Verifier's selected responses ($U_v$) on the training/test set at each checkpoint.
>
> > I don't see the necessity of formulating everything as differential equations as the LLM self-improvement update is discrete.
>
> Discrete step-by-step updates are often analytically intractable to solve globally. Following established theoretical methodologies in model distillation and self-improvement (Mobahi et al.2020, Allen-Zhu \& Li 2023, Boix-Adsera 2024), we approximate the discrete process with continuous equations. This approach allows us to derive closed-form analytical solutions (Equations 10 \& 11), thereby enabling us to prove that self-improvement follows an exponential convergence law.
>
> > it is unclear to me what TF and QE methods are.
>
> TF (True/False) is a discrete verification method where the verifier acts as a binary classifier (Score = $0$ or $1$). The solver generates $N$ responses, denoted $\hat{y}\_{i,1},\cdots,\hat{y}\_{i,N}$, for a prompt $x_i$. The verifier is then tasked with answering whether each response $\hat{y}\_{i,j}$ is correct. If the verifier deems a response $\hat{y}\_{i,j}$ correct, its score $s(\hat{y}\_{i,j})$ is set to $1$; otherwise, it is set to $0$;
> QE (Quality Evaluation) is a continuous verification method where the verifier assigns a scalar quality score $s\in[0,1]$ to a response. The solver generates $N$ responses, $\hat{y}\_{i,1},\cdots,\hat{y}\_{i,N}$, for a prompt $x\_i$. The verifier then assigns a continuous score $s(\hat{y}\_{i,j})$ between 0 and 1 to each response based on its quality. A score of $s(\hat{y}\_{i,j})=0$ indicates a completely incorrect answer, while $s(\hat{y}\_{i,j})=1$ indicates a completely correct answer. These are detailed in Appendix D.2. We are eager to provide more details to help your evaluation.
>
> > The result of the gap narrows during self-improvement does not seem new to me; it appeared in many previous works already. The paper needs to highlight what are the significance of the current results.
>
> We sincerely thank the reviewer for this opportunity to further clarify the core contribution and novelty of our work.
> While we acknowledge that the phenomenon of the gap narrowing has been empirically observed in prior studies, the novelty of this paper lies not in the observation itself, but in mathematically characterizing the training dynamics via the concept of Potential Energy. Previous works established the existence of the gap. They did not model how this gap evolves over time. Our work fills this void by formulating the gap as a Potential Energy ($E(t)$) that drives the optimization process. By linking the gap to the rate of change ($dG/dt$) through differential equations, we transform a static observation into a dynamic mathematical model. This energy-based framework allows us to derive the specific exponential convergence law governing the training curve.
> The significance of the current results lies in predictability and strategy optimization. Corollary 3.2 allows to calculate the exact number of epochs required to reach a target accuracy using only early-stage data, preventing compute waste. Our theoretical analysis in Section 5 (Cross-Improvement) proves that the total amount of external data matters more than the timing of its injection. This simplifies large-scale data pipeline engineering.

---

> > ### Author Response · Authors · 2025-11-23
> > **Clarifications (2)**
> >
> > > Can the authors show if this is even possible in any toy settings?
> >
> > We thank the reviewer for pointing it out!
> > Indeed, this linearity holds as a first-order approximation in a simplified setting.
> > Consider a Bernoulli policy parametrized by success probability $p(t)$. Let the Solver capability be related to the negative log-probability: $U_s \approx -\log p$. Let the Verifier capability be a fixed target (ground truth): $U_v \approx 0$ (representing perfect certainty). The Gap is defined as: $G(t) = U_s - U_v = -\log p$.
> >
> > We further assume that the rate of change in probability $dp/dt$ follows $\frac{dp}{dt} \propto -p \log p$.
> > This indicates that the model approximately converges when $p \to 0$ and $p \to 1$, where $\frac{dp}{dt} \to 0$.
> >
> >
> > We then derive the rate of change for $U_s$ using the chain rule:
> > $$
> > \frac{dU_s}{dt} = \frac{d(-\ln p)}{dt} = -\frac{1}{p} \frac{dp}{dt} \propto \log p = -G
> > $$
> >
> > This matches the form of our Equation (1): $\frac{dU_s}{dt} = -\alpha E(t)$, implies that $E(t) \propto G(t)$.
> > This provides evidence on constructing the linear assumption between $E$ and $G$, with a toy example.
> >
> >
> > > Can the authors explain why does eq 12. a reasoning (reasonable?) model?
> >
> >  We interpret the question as inquiring about the reasonableness of the mathematical formulation in Equation 12. We justify its validity from both theoretical and empirical perspectives:
> >
> > 1. **Theoretical Rationality:** Equation 12 models the capability enhancement as a gain factor mechanism. The term $(1+\gamma\eta_t)^{-1}$ acts as a discount factor on the uncertainty. As the ratio of high-quality external data $\eta_t$ increases, the verifier's uncertainty $U_v$ decreases monotonically. This phenomenological formula captures the intuition that external data reduce the system's entropy, with $\gamma$ representing the efficiency of this external information. From a technical standpoint, we explicitly chose this functional form for its mathematical tractability. This specific inverse-linear form allows the coupled dynamics to be approximated by linear matrix operations. If we were to use more complex non-linear scaling laws, the system would become analytically intractable, preventing the derivation of the closed-form approximate solution in Proposition 5.1. This form strikes a necessary balance between capturing physical intuition and maintaining solvability.
> >
> > 2. **Empirical Validation:** Crucially, the validity of this modeling assumption is supported by our experimental results. Based on Eq.12, we derived Proposition 5.1, which predicts that the final solver capability depends primarily on the total amount of external data, rather than its specific allocation timing. Our experiments in Section 5.2 (Table 1) confirm this conclusion. We observed that different allocation strategies (Early, Uniform, Late) yielded negligible differences in final accuracy. This alignment between the theoretical prediction derived from Eq.12 and the empirical outcome strongly validates the reasonableness of the model.
> >
> > > In the final sentence of section 5.2, how did we reach the conclusion that the small difference in accuracy validates prop. 5.1?
> >
> > We respectfully clarify that the experiment validates the conclusions derived from Proposition 5.1, rather than the proposition 5.1. Proposition 5.1 provides the approximate solution for the final uncertainty $U(T)$. As stated in the text following the proposition, a key theoretical conclusion derived from this formula is that the final solver capability depends primarily on the summation of external data usage $\sum_{t=1}^{T}\eta_{t}$, rather than the specific timing ($\eta_t$) of its introduction. In Table 1, we compared three distinct allocation strategies (Early, Uniform, Late) while keeping the total external data budget ($\sum \eta_t$) constant. The results showed that the difference in final accuracy between these diverse strategies was slight. This empirical insensitivity to timing aligns perfectly with "the timing of using external data is not crucial", thereby validating the conclusions under the framework of Proposition 5.1.
> >
> > We sincerely thank you again for your constructive suggestions, which have been instrumental in improving the quality and clarity of our work. We have carefully addressed your questions in the responses above and incorporated the corresponding revisions into the updated manuscript (highlighted in blue). We are eager to provide any further clarifications to help your evaluation!

---

### Official Review · Reviewer_nJpU · 2025-11-01

**Soundness:** 3
**Presentation:** 3
**Contribution:** 3
**Rating:** 4
**Confidence:** 3

**Summary:**

This paper develops a theoretical framework to explain how LLMs improve themselves without external data. It introduces the solver–verifier gap, where the solver represents generation ability and the verifier represents evaluation ability. Using coupled differential equations inspired by potential energy, the authors model the training dynamics and show that solver and verifier capabilities converge exponentially.

**Strengths:**

1. First clear formalization of LLM self-improvement dynamics through solver–verifier interactions.
2. The differential-equation approach captures exponential convergence behavior that aligns well with empirical trends.
3. Experiments across datasets and LLMs demonstrate strong fits (R² > 0.9) to the theoretical model, reinforcing its predictive power.
4. The extension to limited external data regimes adds practical insight into data allocation strategies.

**Weaknesses:**

1. The model is phenomenological rather than derived from first principles; it lacks a deep mechanistic explanation of why solver-verifier dynamics follow this form.
2. Assumes linear potential energy and time-invariant coefficients (α, β), which may not generalize to all LLM training settings.
3. The connection between theoretical variables (E(t), G(t)) and practical LLM learning signals remains abstract.
4. The paper omits discussion of computational cost, convergence rate sensitivity, and implications for large-scale training.

**Questions:**

See weakness above.

---

> ### Author Response · Authors · 2025-11-23
> **Clarifications (1)**
>
> We sincerely thank you for the detailed and constructive review. We are greatly encouraged that you recognized the core strengths of our work. We also deeply value your critical comments regarding the theoretical foundations and practical implications, which have guided us to clarify the physical meaning of our variables and add actionable insights for large-scale training. We address your concerns point-by-point below.
>
> >The model is phenomenological rather than derived from first principles; it lacks a deep mechanistic explanation of why solver-verifier dynamics follow this form.
>
> We agree that our model is phenomenological. We respectfully argue that this is a methodological strength when studying behaviors in complex systems like LLMs, rather than a weakness. Our approach is analogous to Thermodynamics, which describes the macroscopic behavior of gases (e.g., $PV=nRT$) without modeling the first principles mechanics of every individual molecule. Similarly, modeling the gradient dynamics of billions of parameters is currently intractable. Our goal is to capture the macroscopic dynamics of self-improvement. Our model demonstrates a strong fit ($R^2>0.9$) (Figures 2, 4, 7) across different datasets and models. This strongly suggests that our phenomenological model, though simple, effectively captures the true underlying dynamics of the system.
>
> >Assumes linear potential energy and time-invariant coefficients ($\alpha, \beta$), which may not generalize to all LLM training settings.
>
> We clarify that linearity is inspired by empirical observations, not just a prior assumption, and the time-invariance is a local approximation. Figures 2 and 7 plot the $dG/dt$ against $G(t)$, revealing a linear relationship between the potential energy $E(t)$ and gap $G(t)$ across various models.
> Regarding the coefficients ($\alpha,\beta$), while they may evolve in entirely different training paradigms, modeling them as time-varying functions would result in complex modeling processes and make the parameter estimation nearly impossible without further assumptions. Despite this simplification, the strong empirical fit ($R^2>0.9$) observed in our experiments (Figures 4 \& 11) demonstrates that the constant-coefficient assumption is sufficient to capture the macroscopic training dynamics with high precision.
>
> >The connection between theoretical variables ($E(t),G(t)$) and practical LLM learning signals remains abstract.
>
> We apologize for the lack of clarity. The variables in our phenomenological model map directly to concrete, measurable quantities in the LLM training process:
> 1. **The capability gap $G(t)$:** As defined in $Eq.(6),G(t)=U_v(t)-U_s(t)$. This represents the "Quality Delta" between the model's current generation performance $U_s$, and the quality of the verification output $U_v$. In practice, this is measured by comparing the average NLL of the model's outputs against the average NLL of the Best-of-N responses selected for training.
> 2. **The potential energy $E(t)$:** E(t) is a function of the gap, approximated linearly as $E(t)=kG(t)-b$. This represents the power which drives self-improvement. Equation $8$ states $\frac{dU_s(t)}{dt}=-\alpha E(t)$. This means $E(t)$ maps directly to the speed of improvement of the model. In a practical training run, a high $E(t)$ indicates that the gap is large enough to provide strong gradients, leading to a rapid drop in loss. A low $E(t)$ indicates the model has exhausted the information in the current gap, leading to a training plateau.
>
> Therefore, monitoring $E(t)$ and $G(t)$ might be equivalent to monitoring the efficiency of the data relative to the model's current state.

---

> > ### Author Response · Authors · 2025-11-23
> > **Clarifications (2)**
> >
> > >The paper omits discussion of computational cost, convergence rate sensitivity, and implications for large-scale training.
> >
> > We thank the reviewer for highlighting these practical dimensions. Our theoretical framework actually yields mathematically derived implications for optimizing computational costs and training strategies in large-scale regimes. We clarify these based on Corollary 3.2 and Section 5. Corollary 3.2 explicitly derives the relationship between the required training epochs $t$ and the desired convergence tolerance $\epsilon$:
> > $t>\frac{ln(\delta/\epsilon)}{k(\alpha-\beta)}$.
> >
> > In a large-scale setting, one can perform a few epochs to estimate the system parameters ($k,\alpha,\beta$). Using these, one can analytically predict the total epochs $t$ needed to reach a target performance. This allows to assess whether a training run will be cost-effective before committing full resources. In large-scale training, incorporating high-quality external data (Cross-Improvement) is a common but expensive practice. Our framework provides theoretical guidance on how to allocate this budget. Our analysis in Section 5 (Proposition 5.1) proves that the final Solver capability depends primarily on the total amount of external data used, rather than the specific timing of its introduction. This significantly simplifies the engineering of large-scale data pipelines. It suggests that complex strategies for mixing synthetic and external data might be unnecessary regarding the final performance.
> >
> > We sincerely thank you again for your constructive suggestions, which have been instrumental in improving the quality and clarity of our work. We have carefully addressed your questions in the responses above and incorporated the corresponding revisions into the updated manuscript (highlighted in blue). We are eager to provide any further clarifications to help your evaluation!

---

### Author Response · Authors · 2025-12-01
**General response**

We sincerely thank the Area Chair and all reviewers (**Sog4**, **njpu**, **zspe**, **v9jn**) for their constructive feedback.
We are encouraged that the reviewers value our approach to modeling complex self-improvement dynamics using simplified differential equations. In response to the insightful questions regarding **generalization across domains**, **validity of assumptions**, and **upstream training confounders**, we have conducted extensive additional experiments.

During the rebuttal phase, we have actively participated in the discussion to ensure all concerns were fully addressed. We are particularly grateful to **Reviewer Sog4**, who confirmed that our responses resolved his/her questions and explicitly expressed a willingness to raise his/her score. While we regret not receiving further feedback from the other reviewers regarding our detailed responses and new results, we hope that the extensive additional experiments and theoretical clarifications presented below fully address their initial comments and merit a re-evaluation of our work.

Below is a summary of our major updates and new experimental results:

1. **New Experiments: Generalization and Validation**

To address concerns about the scope of our framework and the validity of our assumptions, we have added the following experiments (detailed in the updated Appendix):

*Generalization to Code and Logic (Addressing Reviewer Sog4):*  We expanded our evaluation beyond mathematical reasoning to include Code Generation (MBPP) and Logical Reasoning (ProntoQA). The training dynamics on these distinct tasks strictly follow the exponential convergence law proposed in our framework ($R^2 > 0.9$), proving that our model generalizes well beyond math tasks (see **Figures 10, 11, 14, 15** in Appendix D).

*Controlled Upstream Comparison (Addressing Reviewer v9jn):*
To disentangle the impact of pre-training stages, we conducted controlled experiments using the **EvoLM** suite. We compared self-improvement dynamics across three distinct stages: **SFT Baseline**, **Mid-trained**, and **Post-trained** models. We identified distinct dynamical regimes. Younger models (Base/Mid) exhibit higher plasticity (higher learning rates), while Post-trained models show flattened trajectories. This confirms our framework effectively captures how model maturity diminishes the potential for self-improvement (see **Figures 22 \& 23**).

*Impact of Model Scale (Addressing Reviewer v9jn):* We compared EvoLM-1B vs. EvoLM-4B to analyze the effect of FLOPs. Larger models exhibit a higher exponential decay rate ($\alpha$) and lower asymptotic uncertainty, validating that scale creates more "optimization space" within our theoretical model.

*Statistical Validation of Memoryless Dynamics (Addressing Reviewer Sog4):*
To justify modeling the dynamics as a conservative vector field (driven by current state rather than history), we performed a **Partial F-test** comparing a model with current gap $G(t)$ versus one including history $G(t-1)$. The test confirmed that adding historical state information does not significantly improve the model fit, statistically validating our Potential Energy assumption.

2. **Theoretical Clarifications**

*Validity of Uncertainty as a Metric (Addressing Reviewer Sog4):* We provided empirical evidence showing a strong negative correlation ($|r| > 0.8$) between our uncertainty metric ($U_s$) and task accuracy. We further clarified that minimizing uncertainty is mathematically equivalent to the sharpening mechanism inherent in SFT.

*Phenomenological Nature (Addressing Reviewers njpu \& zspe):* We clarified that our linear and time-invariant assumptions are **phenomenological approximations** (akin to thermodynamics) rather than first-principles derivations. The consistently high coefficient of determination ($R^2 > 0.9$) across diverse models and datasets justifies this abstraction as a powerful tool for capturing macroscopic training dynamics.

*Practical Implications for Large-Scale Training:*
We highlighted that **Corollary 3.2** allows practitioners to predict the total epochs required for convergence using only early-stage data, offering a concrete way to optimize compute budgets. Our theoretical analysis on **Cross-Improvement** (Proposition 5.1) and supporting experiments (Table 1) demonstrate that the **total amount** of external data matters significantly more than the **timing** of its injection, simplifying pipeline engineering for large-scale training.

We believe these revisions and additional experiments have significantly strengthened the robustness and generalizability of our work. We thank the reviewers again for guiding these improvements.

---

### Meta-Review · Area_Chair_Q2An · 2026-01-10

**Summary:**

The reviewers provided some good comments for the authors, and authors have also shown genuine interests in engaging reviewers with additional experiments and detailed experimentation.

I don’t see any major items left for this work. Hence I recommend an accept (poster) for this paper.

**Reviewer Concerns:**

The reviewers most comments are about the modeling approach of this work and the clarity of some explanations. The authors have provided reasonable explanations for them.

No major outstanding issues found.

**Reviewer Scores:**

Reviewer Sog4 would have raised his score for sure. The other two reviewers (ZsPe, nJpU) may also raise their 4 rating

---

### Decision · Program_Chairs · 2026-01-26

Accept (Poster)